# AUC-CL: A Batchsize-Robust Framework for Self-Supervised Contrastive Representation Learning

**Rohan Sharma**[1], **Kaiyi Ji**[1], **Zhiqiang Xu**[2], **Changyou Chen**[1]
[1]University at Buffalo, [2]MBZUAI
`{rohanjag, kaiyiji, changyou}@buffalo.edu, {zhiqiang.xu}@mbzuai.ac.ae`

## Abstract

Self-supervised learning through contrastive representations is an emergent and promising avenue, aiming at alleviating the availability of labeled data. Recent research in the field also demonstrates its viability for several downstream tasks, henceforth leading to works that implement the contrastive principle through innovative loss functions and methods. However, despite achieving impressive progress, most methods depend on prohibitively large batch sizes and compute requirements for good performance. In this work, we propose the **AUC-C**ontrastive **L**earning, a new approach to contrastive learning that demonstrates robust and competitive performance in compute-limited regimes. We propose to incorporate the contrastive objective within the AUC-maximization framework, by noting that the AUC metric is maximized upon enhancing the probability of the network's binary prediction difference between positive and negative samples which inspires adequate embedding space arrangements in representation learning. Unlike standard contrastive methods, when performing stochastic optimization, our method maintains unbiased stochastic gradients and thus is more robust to batchsizes as opposed to standard stochastic optimization problems. Remarkably, our method with a batch size of 256, outperforms several state-of-the-art methods that may need much larger batch sizes (e.g., 4096), on ImageNet and other standard datasets. Experiments on transfer learning and few-shot learning tasks also demonstrate the downstream viability of our method. Code is available at AUC-CL.

## 1 Introduction

As deep learning continues to pervade across several facets of our livelihood, we come to an increasing realization that the developments in the field have outpaced the availability of data, especially labelled data. An increasing apprehension of these limits over time has led to research towards a data efficient and viable alternative of self-supervised learning (SSL). Recent SSL pretraining based methods (Grill et al., 2020; Chen et al., 2020; Caron et al., 2020; Chen et al., 2021; Lee et al., 2021; Li et al., 2021; Tomasev et al., 2022) have successfully recovered the performance of and often outperformed supervised methods for the downstream tasks on classification, object detection, recognition, and segmentation, to name a few, in the low data regime. Large scale language models such as BERT (Devlin et al., 2018), RoBERTa (Liu et al., 2019), XLM-R (Conneau et al., 2019), and GPT (Brown et al., 2020), which have their roots in foundational methods such as Word2Vec (Mikolov et al., 2013) and GloVe (Pennington et al., 2014), serve as the exemplars of surprising cognizance of language(s) using the self-supervised pretraining routine.

However, a significant drawback of these models is the requirement of steep computing capacities and training times, largely due to the nature of the contrastive loss function. Several of the aforementioned methods leverage the ubiquitous and ergonomic cross-entropy function as the main objective for their training process. The objective function, albeit robust, often leads to an optimization bias in the computation of the gradient. For example, Chen et al. (2022) illustrates this phenomenon using a simple synthetic experiment, showing that mere usage of SGD based optimization can lead to several sub-optimal solutions. More specifically, as shown by Yuan et al. (2022), the NT-Xent objective (a variant of the cross-entropy loss) used in SimCLR (Chen et al., 2020) and several other works can

lead to considerable optimization error: "*SimCLR suffers an optimization error at least in the order of $O(1/\sqrt{B})$ for the objective's gradient norm. Even with $T \to \infty$, its optimization error is always dominated by $O(1/\sqrt{B})$.*" This bias in the updates largely explains the degradation of performance in SimCLR and related methods with smaller batch sizes, which notably arises from the denominator of the contrastive objective (Chen et al., 2022).

In addition to the aforementioned works, Yeh et al. (2022); Chuang et al. (2020) also unambiguously indicate an anomaly specific to the denominator of the contrastive loss. This leads to the conclusion that, in expectation, the standard contrastive loss function is not an unbiased estimate of the true objective, which actually encompasses infinitely many negative samples for each positive one. This motivates us to study alternatives to overcome the intrinsic limitation of contrastive learning. To this end, we propose a new contrastive loss by incorporating the "contrastive" idea into the AUC-maximization framework. Specifically, since AUC only specifies two classes in the definition, we propose to treat the two classes as positive and negative classes for each data point. These positive and negative representations are then optimized through the AUC criterion, *i.e.*, to maximize the area under the ROC curve for positive and negative classes. By optimizing the new loss, one can not only seamlessly model the positive-negative sample interactions as done in standard contrastive learning, but also enjoy merits inherited from the AUC loss, *e.g.*, Yuan et al. (2020) demonstrates the AUC loss is more robust to noise compared to the cross-entropy loss. In addition, we show through an extensive analysis that our loss function provides batch size independent gradient updates and assures us of provable convergence under mild assumptions. In summary, our major contributions are as follows:

- We re-examine some limitations of the standard contrastive learning framework, and propose a new contrastive loss that incorporates contrastive representation within the AUC maximization framework. Our framework comes with nice theoretical properties and inspires superior performance under low batch size regimes.

- We thoroughly analyze the function and its gradient updates and show that our function leads to unbiased gradient updates. Through extensive experiments, we demonstrate the robustness of our framework to small batch sizes.

- We empirically demonstrate that our method outperforms several SOTA methods in SSL, as well as in some popular few-shot learning and transfer learning benchmarks.

## 2 Preliminaries and Related Work

### 2.1 Self-supervised learning

Self-supervision in machine learning has become increasingly promising due to the alleviation of the requirement for labelled data. Successful applications in downstream tasks further reflect well on its merits. The training mechanisms take varying formats that leverage several forms of co-occurrences. Temporally relevant information is utilized in works such as Wang & Gupta (2015); Logeswaran & Lee (2018); Singh et al. (2021); Pan et al. (2021); Gao et al. (2022). For vision based tasks, the highly popular method of using multiple image augmentations towards encouraging feature space separation is leveraged by several works (Dosovitskiy et al., 2014; Bachman et al., 2019; Tian et al., 2020; Gao et al., 2021). Methods that implement alternative contrastive objectives along feature dimensions include SimSiam (Chen & He, 2021), BYOL (Grill et al., 2020), DINO (Caron et al., 2021), VICReg (Bardes et al., 2021), Barlow Twins (Zbontar et al., 2021), and HaoChen et al. (2021) which uses a spectral contrastive objective.

Recent innovations falling under image sample contrastive methods that we consider relevant to this work include the SimCLR (Chen et al., 2020), MoCo v3 (Chen et al., 2021), CLIP (Radford et al., 2021), DCL (Yeh et al., 2022) and others, which utilize variants of the cross-entropy criterion. A prominent characteristic of these works is the requirement for large batch sizes in order to enhance performance, which often mandates a steep compute requirement and training time.

### 2.2 Contrastive Learning and its Limitations

Contrastive Learning can date back to work in the early 90's (Becker & Hinton, 1992; Bromley et al., 1993), based on the motive of utilization of internally derived teaching signals in a neural

network towards memory and data efficient systems. The core implementation has prevailed to the recent times, which is based on minimization of the distance between features generated from images/modalities belonging to a unique entity. In recent times, (Chen et al., 2020; 2021; Caron et al., 2021; Chen & He, 2021), the field has successfully leveraged data augmentation strategies towards training large networks that often rival the supervised paradigm. In its most elementary form, the augmented versions of similar samples are drawn closer in the feature space and the dissimilar ones pushed apart. However, owing to the lack of labels, these methods suffer from drawbacks that concern efficient sampling of the negatives.

Standard contrastive learning such as NTXent (Chen et al., 2020) and InfoNCE (Oord et al., 2018) use a variant of the cross entropy as the loss function. For a batch $\mathcal{B}$ (containing the indices of the data samples) and the corresponding features vectors $\{z_i\}$, the NTXent loss is formulated as

$$\mathcal{L}^{\text{NTXent}} = -\frac{1}{n} \sum_{i,j \in \mathbb{B}} \log \frac{\exp(\text{sim}(z_i, z_j)/\tau)}{\sum_{k=1:k \neq i}^{2n} \exp(\text{sim}(z_i, z_k)/\tau)} \tag{1}$$

where $\text{sim}(u, v)$ denotes the cosine similarity of features $u$ and $v$; $\tau$ is the temperature parameter and $n = |\mathcal{B}|$. Of particular interest to this work is the denominator $g(\mathbf{w}, i) = \sum_{k=1:k \neq i}^{2n} \exp(\text{sim}(z_i, z_k)/\tau)$, where $\mathbf{w}$ refers to the parameters of the network. This takes the form of a sum-exponent of similarities that ideally correspond to the negative class. However, the gradient estimate of this function with minibatches, given by

$$\tau \nabla_{\mathbf{w}} \mathcal{L}^{\text{NTXent}} = -\frac{1}{n} \nabla_{\mathbf{w}} \text{sim}(z_i, z_j) + \frac{1}{n} \frac{\nabla g(\mathbf{w}, i)}{g(\mathbf{w}, i)} \, , \tag{2}$$

is a biased estimator of the true gradient (Chen et al., 2022). In practice, large batch sizes are needed to trade off the accuracy and computational efficiency.

There have been some efforts made to mitigate this dependence from several perspectives. For instance, Yeh et al. (2022) construct a decoupled objective that discards the presence of positive samples in the InfoNCE criterion and demonstrate gains. Chuang et al. (2020) also remedy the presence of positive samples in the denominator by constructing an estimator for bias correction. Ge et al. (2021), Kalantidis et al. (2020) and Robinson et al. (2020) attempt negative mining to improve performance without explicitly addressing the bias issue.

To our knowledge, very little work has explicitly addressed the bias phenomenon and suggested solutions. Chen et al. (2022) demonstrate the bias phenomenon using a simple synthetic experiment and propose an expectation maximization based algorithm which adaptively weighs the negative samples in the minibatch. Yuan et al. (2022) quantify the bias in the order of $O(1/\sqrt{B})$ and suggest a solution that maintains track of the denominator terms and uses its moving average to compute the gradient at every step which comes at a cost to the memory and computation. Although these works provide remedies, the principle working mechanism is based on an asymptotically diminishing bias. We emphasise that our method not only is a simplistic principled approach, it is also provably free of the bias and does not rely on momentum encoders, stop-gradient methods, negative mining or any additional tricks for better performance.

## 2.3 AUC MAXIMIZATION

Area under the ROC curve (AUC) is one practical metric for evaluation of model performance, which accounts for imbalances in dataset classes and overcomes the shortcomings of metrics such as accuracy. In principle, maximization of AUC leads towards enhancement of the prediction score of positive class, relative to the other negative classes. Some works that have studied pairwise maximization methods towards this objective are Gao & Zhou (2015), Joachims (2005), Herschtal & Raskutti (2004). However, a natural challenge to the optimization process for AUC that arises as a consequence is the high computational complexity ($O(N^2)$) which results from directly maximizing the positive class scores against the paired negative samples which inhibits learning from large-scale data. Ying et al. (2016) proposed an equivalence that allows for a min-max optimization objective that alleviates the $O(N^2)$ complexity of the objective and allows for an efficient optimization objective. Thereafter, improvements over this work were researched by Natole et al. (2018), Liu et al. (2018)

and Yuan et al. (2020). Notably, Yuan et al. (2020) address laxities of the aforementioned methods, namely the adverse effects of easily classifiable data and the sensitivity to noisily labelled data, through a margin based component in the loss function. The works above inspired us to construct our objective function that proves to be a prominent option for SSL based tasks that is very reliable in limited compute related circumstances due to its robustness to batch size.

# 3   THE PROPOSED METHOD

We first discuss some work of standard AUC metric maximisation frameworks, which was originally used for performance evaluation of classifiers. We then describe the motivation behind our formulation from the perspective of these approaches and subsequently, we propose our new framework for batchsize-robust contrastive representation learning by incorporating the contrastive idea into the AUC maximization formulation; and conduct a theoretical analysis to show that the gradient updates resulting from our framework are unbiased and therefore robust to batch size variance. The convergence theory for our method is also developed.

## 3.1   NOTATIONS

Let $\mathcal{D} = \{\mathbf{x}_1, ..., \mathbf{x}_n\}$ denote the set of training images and let $\mathcal{P}$ denote the complete set of data augmentations with a randomly picked data augmentation denoted by $\mathcal{A}(.) \in \mathcal{P}$. For a randomly selected sample $\mathbf{x} \in \mathcal{D}$, consider the set $\mathcal{M}_i$ defined as $\mathcal{M}_i = \{j : \mathcal{A}(\mathbf{x}_j) \text{ for } \forall \mathcal{A} \in P, \forall \mathbf{x}_j \in \mathcal{D} \backslash \{\mathbf{x}_i\}\}$, which denotes the set of indices of all samples and their augmentations excluding those of $\mathbf{x}_i$. Analogous to this, for a batch $\mathcal{B}$, $\mathcal{B}_i = \{j : \mathcal{A}(\mathbf{x}_j) \text{ for } \forall \mathcal{A} \in P, \forall \mathbf{x}_j \in \mathcal{B} \backslash \{\mathbf{x}_i\}\}$. Let $E(.)$ define the encoder network comprising of the backbone network $f_{\mathbf{w}}$ and the projector network $g_{\mathbf{w}}$, used for the generation of feature vectors for every image. Upon completion of training, we discard the projector $g_{\mathbf{w}}$. We denote the cosine similarity between two encoded samples $u = E(\mathbf{x}_u)$ and $v = E(\mathbf{x}_v)$ by $\text{sim}(u, v)$ defined by $\text{sim}(u, v) = \frac{u^t v}{||u|| \times ||v||}$.

## 3.2   AUC MAXIMIZATION

We start by a brief review of the AUC maximization framework, which, as a problem for deep learning, is best formulated with a differentiable form. This can facilitate learning by application of backpropagation. Specifically, the AUC metric can be formulated using the Wilcoxon-Mann-Whitney statistics (Mann & Whitney, 1947; Wilcoxon, 1992) as:

**Definition 3.1.** Given a classifier $h_{\mathbf{w}}$ parameterized by $\mathbf{w}$ and labeled samples $\{\mathbf{x}, \mathbf{x}' | y = 1, y' = -1\}$, the area under the receiver operator characteristic curve is defined by,

$$\text{AUC}(w) = \text{P}(h_{\mathbf{w}}(\mathbf{x}) \geq h_{\mathbf{w}}(\mathbf{x}') \mid y = 1, y' = -1). \tag{3}$$

This formulation leads to an $O(N^2)$ complexity due to the explicit requirement of positive-negative pairs for optimization. A widely used reformulation by introducing the square surrogate loss lends to an ease of optimization and alleviates the high complexity (Ying et al., 2016; Natole et al., 2018; Liu et al., 2018; Yuan et al., 2020), given by

$$L(\mathbf{w}) = \mathbb{E}[(1 - h_{\mathbf{w}}(\mathbf{x}) + h_{\mathbf{w}}(\mathbf{x}'))^2 \mid y = 1, y' = -1]. \tag{4}$$

To make the problem in 4 more tractable, we use the following equivalent reformulation theorem (Yuan et al., 2020).

**Theorem 3.2.** *Minimising (4) is equivalent to the following min-max problem:*

$$\min_{\mathbf{w} \in \mathbf{R}^\mathbf{d}} L(\mathbf{w}) = \min_{\mathbf{w} \in \mathbf{R}^\mathbf{d}} \max_{\alpha \in \mathbb{R}} \mathbb{E}[L\prime(\mathbf{w}, \alpha)], \text{ where}$$

$$L\prime(\mathbf{w}, \alpha) \triangleq \left[(h_{\mathbf{w}}(\mathbf{x}) - a(\mathbf{w}))^2 \mid y = 1\right] + \left[(h_{\mathbf{w}}(\mathbf{x}') - b(\mathbf{w}))^2 \mid y' = -1\right]$$
$$+ \max_{\alpha} \left\{2\alpha(1 - a(\mathbf{w}) + b(\mathbf{w})) - \alpha^2\right\}. \tag{5}$$

*The optimal values of $a, b, \alpha$ given $\mathbf{w}$ are $a = a(\mathbf{w}) := \mathbb{E}[h_{\mathbf{w}}(\mathbf{x}) \mid y = 1]$, $b = b(\mathbf{w}) := \mathbb{E}[h_{\mathbf{w}}(\mathbf{x}) \mid y = -1]$ and $\alpha = 1 + b - a$, respectively.*

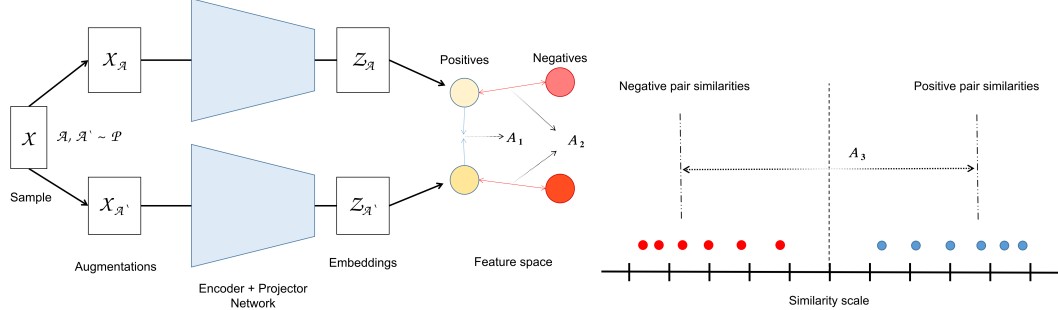

Figure 1: **The proposed AUC-CL framework:** The component $A_1$ serves to enhance the proximity of positive pairs whereas $A_2$ enhances the distance between positive-negative pairs. Additionally, $A_3$, governed by the non-zero $\alpha$ parameter augments the gap between the expected values of similarities of positive and negative pairs, thus preventing collapse.

### 3.3 THE PROPOSED AUC-CL FRAMEWORK

We describe the incorporation of the contrastive idea into the AUC maximization framework. In our setting, we adopt the cosine similarity metric to evaluate the feature space proximity of the data samples. Given a sample $\mathbf{x}$, we generate feature vectors from two random augmentations of the sample, $E(\mathbf{x}_u)$ and $E(\mathbf{x}_v)$. Thus, for a batch of size $|\mathcal{B}|$ we derive two sets of augmented features, each of size $|\mathcal{B}|$. We then evaluate the cosine similarity matrix using the two sets, which forms the matrix of scores that we eventually use in our objective function.

Note both the contrastive loss and the AUC formula contain positive and negative data, yet their definitions are different. Therefore, one needs to make transformations between them in order to achieve incorporation of contrastive loss into the AUC formula. To this end, we propose that each positive and negative classes in the AUC formula should correspond to a pair of data samples in our setting. Specifically, we consider a score $\text{sim}(i, j)$ between samples $\mathbf{x}_i$ and $\mathbf{x}_j$, each subjected to random augmentation operators $\mathcal{A}$ and $\mathcal{A}'$ respectively, where $(i \in [1, |\mathcal{B}|], j \in [1, |\mathcal{B}|])$ belong to the positive class in the AUC formula if it is derived from augmentations of the same source image, *i.e.*, $i = j$, and is denoted as $y_{ii+} = 1$. Similarly, it is called negative class (denoted as $(y_{ij} = -1)$) if the pair is derived from separate images $i \neq j$. Based on this, the original AUC maximization objective in (5) is generalized to what we coin as the AUC contrastive learning:

$$\min_{\mathbf{w} \in \mathbf{R}^\mathbf{d}} \max_{\alpha \in \mathbb{R}} \mathbb{E}_{\mathbf{x}_i \sim \mathcal{D}, \mathcal{A}, \mathcal{A}' \sim \mathcal{P}}[L'_s(\mathbf{w}, \alpha; \mathbf{x}_i, \mathcal{A}, \mathcal{A}', \mathcal{B}_i)] \tag{6}$$

where the loss $L'_s(\mathbf{w}, \alpha; \mathbf{x}_i, \mathcal{A}, \mathcal{A}', \mathcal{B}_i)$ at data $\mathbf{x}_i$ and two data augmentations $\mathcal{A}$ and $\mathcal{A}'$ are given by

$$L'_s(\mathbf{w}, \alpha; \mathbf{x}_i, \mathcal{A}, \mathcal{A}', \mathcal{B}_i) = \left[(1 - \text{sim}(i, i+) + \sum_{j \in \mathcal{B}_i} \text{sim}(i, j))^2 \mid y_{ii+}, y_{ij}\right] \tag{7}$$

$$= \underbrace{\left[(\text{sim}(i, i+) - a(\mathbf{w}))^2 \mid y_{ii+}\right]}_{a_1(\mathbf{w})} + \underbrace{\sum_{j \in \mathcal{B}_i} \left[(\text{sim}(i, j) - b(\mathbf{w}))^2 \mid y_{ij}\right]}_{a_2(\mathbf{w})} + \underbrace{\max_{\alpha} \left\{2\alpha(1 - a(\mathbf{w}) + b(\mathbf{w})) - \alpha^2\right\}}_{a_3(\mathbf{w})}.$$

Here, we evaluate the component $a_1(\mathbf{w})$ for the positive class and $a_2(\mathbf{w})$ for the negative class as the notation suggests. Inspired by the conventional contrastive learning objectives, we take a sum over the negative terms $y_{ij}$ as a measure to enhance the impact of the negative sample pairs, which in turn can improve the similarity scores for positive sample pairs. This leads to a significant improvement in the performance.

**A practical reformulation** We propose a reformulation to the original problem in equation 6 to improve practical performance. 1) First, according to Theorem 3.2, the optimal values for $a(\mathbf{w})$ and $b(\mathbf{w})$ are: $a(\mathbf{w}) := \mathbb{E}[\text{sim}(i, i+) \mid y_{ii+}], b(\mathbf{w}) := \mathbb{E}[\sum_{j \in \mathcal{B}_i} [\text{sim}(i, j) \mid y_{ij}]$. We can substitute these optimal values into equation 6 and directly optimize over the encoder parameter $\mathbf{w}$. This could lead to better convergence as one does not need to alternate between optimizing $a$ and $\mathbf{w}$. However, to avoid approximation errors, the expectations need to be exactly evaluated. In practice, only stochastic

approximations calculated from mini-batches are available. Thus, replacing $a(\mathbf{w})$ and $b(\mathbf{w})$ with their stochastic estimations would induce bias if the evaluation function is non-linear, *e.g.*, $a_1(\mathbf{w})$ and the $a_2(\mathbf{w})$ terms in equation 6. As a result, we only do the substitution for $a(\mathbf{w})$ and $b(\mathbf{w})$ for the linear terms, *e.g.*, the $a_3(\mathbf{w})$ term. 2) Second, once we substitute $a(\mathbf{w})$ and $b(\mathbf{w})$ with their stochastic approximations in term $a_3(\mathbf{w})$, the hyper-parameter $\alpha$ has to be independent w.r.t. $\mathbf{w}$ in order to avoid bias introduced in stochastic approximation. Thus, we simply fix $\alpha$ in the experiments. We conduct ablations with various values of $\alpha$ to illustrate the robustness of our function to the values. Additionally, one may opt to fix $a$ and witness marginal improvements. We conjecture that this may allow ease in optimization for the uniformity task of representation learning, which is non-trivial as compared to the alignment task Wang & Isola (2020). As a result, we propose the following new loss in practice as our objective function:

$$L'_s(\mathbf{w}, b; \mathbf{x}_i, \mathcal{A}, \mathcal{A}', \mathcal{B}_i) = \underbrace{\left[(\text{sim}(i, i+) - a)^2 \mid y_{ii+}\right]}_{A_1} + \underbrace{\sum_{j \in \mathcal{B}_i} \left[(\text{sim}(i, j) - b)^2 \mid y_{ij-}\right]}_{A_2} \quad (8)$$

$$+ \underbrace{\left\{2\alpha\left[1 - \text{sim}(i, i+) \mid y_{ii+} + \sum_{j \in \mathcal{B}_i} \text{sim}(i, j) \mid y_{ij-}\right] - \alpha^2\right\}}_{A_3}$$

Based on the loss, we describe the algorithm for pretraining using our contrastive loss in Algorithm 1. An intuitive illustration of the method and the functions of its components is given in the Figure 1.

### 3.4 CONVERGENCE ANALYSIS

We conduct the analysis of the gradient estimators for various components of our function in the under section A.1 in the Appendix using the standard assumptions of smoothness and continuity. In a high level, we evaluate the gradients of the various components of the Global Contrastive Objective given by equation 8 and the stochastic gradient estimators of these components which are used in batch wise optimization. We then establish the smoothness properties of these components and that the stochastic gradient estimator is an unbiased estimate of the true gradient, using standard assumptions. Subsequently, using these properties, we establish the following convergence result.

**Theorem 3.3.** *Assume the cosine similarity function $\text{sim}(i, j)(\mathbf{w})$ is smooth and Lipschitz continuous w.r.t. $\mathbf{w}$ and $\max\{a, b\} \leq \tau$. Choose the stepsize $\beta = \frac{1}{\sqrt{T}}$. Then the stochastic gradients in our algorithm are unbiased. Furthermore, for the optimization parameters $\mathbf{v} = (\mathbf{w}, b)$, we have the following convergence results*

---

**Algorithm 1** : AUC-CL

**Input:** Dataset $\mathcal{D} = \{\mathbf{x}_1, ..., \mathbf{x}_n\}$

Parameters and networks: $\mathbf{w}, f_\mathbf{w}, g_\mathbf{w}$

1: **for** n=1 to N **do**
2:      Sample a batch $\mathcal{B} \leftarrow \{x_i \sim \mathcal{D}\}_{i=1}^{B}$
3:      **for all** $i \in \{1...B\}$ **do**
4:          $x_i, x_{i+} = \mathcal{A}(x_i), \mathcal{A}'(x_i)$
5:          $z_i, z_{i+} = g_\mathbf{w}(f_\mathbf{w}(x_i)), g_\mathbf{w}(f_\mathbf{w}(x_{i+}))$
6:      **end for**
7:      **for all** $i \in \{1...B\}$ and $j \in \{1...B\}$ **do**
8:          $\text{sim}(i, j) = \frac{z_i^T z_j}{||z_i|| \times ||z_j||}$
9:      **end for**
10:     Evaluate $L'_s$ according to 8
11:     $\mathbf{w} \leftarrow \text{optimizer}(\mathbf{w}, \partial_\mathbf{w} L'_s)$
12: **end for**

**Output:** Encoder $f_\mathbf{w}$

---

$$\mathbb{E}\|\nabla L(\mathbf{v}_t)\|^2 \leq \mathcal{O}\left(\frac{1}{\sqrt{T}}\right),$$

*where $t$ is drawn from $\{0, ...., T - 1\}$ uniformly at random.*

*Remark* 3.4. Theorem 3.3 indicates that our algorithm converges to a local optimum at a rate of $\mathcal{O}\left(\frac{1}{\sqrt{T}}\right)$ w.r.t. the number of updates $T$, which is independent of batch sizes. Whereas for the standard contrastive loss with the cross-entropy related variants, the convergence rate is $\mathcal{O}(1/\sqrt{\eta T} + \sqrt{\eta} + 1/\sqrt{B})$ for the learning rate $\eta$ and batch size $B$ (Yuan et al., 2022). Consequently, this suffers from a convergence error that depends on the batch size $B$, which is at least $\mathcal{O}(1/\sqrt{B})$ at the limit of $T \to \infty$ and $\eta \to 0$.

## 4 EXPERIMENTS

Owing to the adaptable nature of our loss function, we simply adapt the widely existing framework of SimCLR (Chen et al., 2020) and MoCo-v3 (Chen et al., 2021) codebases to AUC-CL. We retain the architectural choices and other configurations and explore different batch sizes. In all the experiments unless otherwise specified, similar to the work in Chen et al. (2021), we set the learning rate to be lr $\times$ BatchSize/256, where lr refers to the base learning rate parameter, a hyperparameter that we experiment with and realize that a stable value is set at lr=1.0e-3 with higher values leading to slower convergence and accuracies lower by $2\%$. Moreover, we adopt a learning rate warmup described by Goyal et al. (2017) for 40 epochs followed by a cosine decay schedule (Loshchilov & Hutter, 2016). Our default optimizer is the AdamW (Ilya et al., 2019).

We follow the standard linear evaluation protocol for our models and report the Top-1 ImageNet validation accuracy of a linear classifier trained over pretrained frozen networks. For ImageNet, we train using a random resized crop strategy by cropping the images to 224x224 and resizing to 256x256 during training and using a center crop and resizing for validation data and train for 90 epochs using SGD optimizer and a batch size of 1024 with a learning rate of 0.1 with a cosine annealing decay schedule and no weight decay. We train our models on four NVIDIA RTX A5000 GPUs with a memory of 24.2 GB per GPU.

**ImageNet** We pretrain and finetune AUC-CL on ImageNet using the ResNet-50 and ViT-Small architectures using a batch size of 256. Additionally, we incorporate a multi-crop strategy of augmentation introduced in SwAV (Caron et al., 2020) for ImageNet with $2 \times 224^2 + 8 \times 96^2$ global and local crops respectively. The results are given in Tables 1 and 2. We reproduce/omit the results for some of the listed methods due to unavailability. Remarkably, our method outperforms several prominent methods that use much larger batch sizes in order to ensure strong performances. We also compare against some of the more recent methods of W-MSE (Ermolov et al., 2021), Zero-CL (Zhang et al., 2022b) and ARB (Zhang et al., 2022a) that are batch size robust. We present supplementary findings utilizing a batch size of 1024 for other methods, while maintaining our batch size at 256 for training over 800 epochs in the Appendix under Table 6.

Table 1: **ImageNet Pretraining (ResNet-50)**. Top-1 accuracy for linear evaluation results are listed. Results marked "*" are reproduced using LightlyAI.

| Method | Batch Size | 100 ep | 400 ep | 800 ep |
|---|---|---|---|---|
| SimCLR | 4096 | 66.5 | 69.8 | 70.1 |
| BYOL | 4096 | 66.5 | 73.2 | – |
| MoCo-v3 | 4096 | 68.9 | 73.1 | 73.8 |
| VICReg | 2048 | 68.6 | 70.6 | 73.2 |
| DINO* | 1024 | 68.3 | 72.9 | – |
| BarlowTwins | 1024 | 67.7 | 73.1 | 73.2 |
| Zero-CL | 1024 | 68.9 | 72.6 | – |
| ARB | 512 | 68.2 | – | – |
| NNCLR+DCL | 512 | – | 71.2 | 74.9 |
| MoCo-v2 | 256 | 67.4 | 71.0 | 72.2 |
| SimCLR+DCL* | 256 | 65.1 | 69.5 | – |
| SimCLR+DCLW* | 256 | 64.5 | 67.8 | – |
| SimSiam | 256 | 68.1 | 70.8 | 71.3 |
| W-MSE | 256 | 69.4 | 72.5 | – |
| AUC-CL | 256 | **69.5** | **73.5** | **75.5** |

Table 2: **ImageNet Pretraining (ViT-S)**. Top-1 accuracy for linear evaluation 10-NN evaluation for $k$-NN are listed after pretraining for 300 epochs.

| Method | Batch Size | $k$-NN | Linear |
|---|---|---|---|
| SimCLR | 1024 | – | 69.0 |
| BYOL | 1024 | 66.6 | 71.4 |
| MoCo-v2 | 1024 | 62.0 | 71.6 |
| MoCo-v3 | 1024 | – | 72.5 |
| MoCo-v3 | 4096 | – | 73.2 |
| SwaV | 1024 | 64.7 | 71.8 |
| AUC-CL | 256 | **70.7** | **73.7** |

**ImageNet-100, Cifar-10 and Cifar-100** We conduct further experiments on standard benchmark image datasets, namely Cifar-10 (Krizhevsky et al., a) and Cifar-100 (Krizhevsky et al., b). Moreover, we also evaluate our function on a smaller scale of the ImageNet (Deng et al., 2009) denoted by ImageNet-100 which consists of a subset of randomly selected 100 classes from the full dataset, in keeping with the protocol followed by Wu et al. (2019) and Yuan et al. (2022). Following general practice, we pretrain using the backbone of ResNet-18 and the set of standard augmentations as given in Chen et al. (2020) and finetune using the same procedure. Table 3 lists our results. We use the popular benchmark given by da Costa et al. (2022) for the other methods. For all the other listed methods, the ImageNet-100 data was trained using a batch size of 128 for 400 epochs and we list our results for the same. The Cifar datasets were trained using a batch size of 256 for 1000 epochs whereas we list our results after pretraining merely for 500 epochs using a batch size of 128. As

listed, AUC-CL outperforms all the listed methods. For the Cifar datasets, a significant margin is gained upon training using merely half the batch size and number of epochs.

Table 3: **Cifar and ImageNet-100 pretraining.** Linear evaluation (Acc@1) results. All methods trained on ImageNet-100 for 400 epochs using a batch size of 256. The Cifar datasets are trained on for 1000 epochs using the same batch size. We train our method on the Cifar datasets for 500 epochs and use a batch size of 128.

| Method | Cifar-10 | Cifar-100 | ImageNet-100 | Method | Cifar-10 | Cifar-100 | ImageNet-100 |
|---|---|---|---|---|---|---|---|
| W-MSE | 88.7 | 61.3 | 67.6 | ARB | 92.2 | 69.6 | 79.5 |
| SwaV | 89.2 | 64.9 | 74.0 | NNCLR | 91.9 | 69.6 | 79.8 |
| SimSiam | 90.5 | 66.0 | 74.5 | MoCo-v3 | 93.1 | 68.8 | 80.4 |
| DINO | 89.5 | 66.8 | 74.8 | BYOL | 92.6 | **70.5** | 80.2 |
| Zero-FCL | 90.5 | **70.2** | 75.7 | Barlow Twins | 92.1 | **70.9** | 80.4 |
| Zero-ICL | 90.5 | 69.3 | 78.0 | AUC-CL | **93.6** | 69.7 | **81.9** |

**Few-Shot Linear Probing**   We further evaluate our ViT-S model through downstream linear probing for classification. We use the ELEVATER benchmark (Li et al., 2022) which conducts 5-shot classification based tasks on 20 public image classification datasets. We run the benchmark with 3 random initializations and use their automated hyperparameter tuning pipeline. Here we compare against other methods that are pretrained on ImageNet but use larger backbones, *e.g.*, ViT-base. We demonstrably perform better than the mentioned methods on several datasets. Notably, our method outperforms on a greater majority of the datasets.

Table 4: **ELEVATER benchmark.** We compare against MoCo-v3 (Chen et al., 2021), DeiT (Touvron et al., 2021) and MAE (He et al., 2022). We list the backbone networks for each. Data-1 to Data-20 correspond to Caltech101, CIFAR10, CIFAR100, Country211, DescriTextures, EuroSAT, FER2013, FGVC Aircraft, Food101, GTSRB, HatefulMemes, KITTI, MNIST, Oxford Flowers, Oxford Pets, PatchCamelyon, Rendered SST2, RESISC45, Stanford Cars and VOC2007 datasets respectively.

| Method | Arch. | Data-1 | Data-2 | Data-3 | Data-4 | Data-5 | Data-6 | Data-7 | Data-8 | Data-9 | Data-10 | **Mean Acc.** |
|---|---|---|---|---|---|---|---|---|---|---|---|---|
| MAE | ViT-B | 59 | 34 | 21.2 | 2.8 | 35 | 64.4 | 21.3 | 7.0 | 7.7 | 17.5 | 33.4 |
| MoCo-v3 | ViT-B | 80.8 | 78.5 | 60.5 | 4.8 | 57.1 | 77.1 | 20.5 | 11.8 | 36.6 | 31.4 | 50.2 |
| DeiT | ViT-B | 86.2 | 70.1 | 61.5 | 4.4 | 52.9 | 62.5 | 14.5 | 24.1 | 41.9 | 46.7 | 54.1 |
| AUC-CL | ViT-S | 87.2 | 76.7 | 61.8 | 4.6 | 57.7 | 77.8 | 19.5 | 28.8 | 38.7 | 49.7 | **54.9** |

| Method | Arch. | Data-11 | Data-12 | Data-13 | Data-14 | Data-15 | Data-16 | Data-17 | Data-18 | Data-19 | Data-20 | **# Wins** |
|---|---|---|---|---|---|---|---|---|---|---|---|---|
| MAE | Vit-B | 51.4 | 46.1 | 63.4 | 50.9 | 17.2 | 54.9 | 50.1 | 38.9 | 6.3 | 18.3 | 2 |
| MoCo-v3 | ViT-B | 50.7 | 46.7 | 64.1 | 79.5 | 76.3 | 54.7 | 50.1 | 61.1 | 13.4 | 47.9 | 2 |
| DeiT | Vit-B | 51.1 | 47.6 | 83.8 | 82.7 | 87.8 | 51.5 | 50.1 | 63.4 | 27.7 | 70.9 | 5 |
| AUC-CL | Vit-S | 50.5 | 51.5 | 77.6 | 86.5 | 76.5 | 62.4 | 50.4 | 65.5 | 24.9 | 49.9 | **11** |

**Other Comparisons**   We compare against another batch size robust method SogCLR (Yuan et al., 2022) using the ResNet-50 backbone and training on ImageNet and ImageNet-100 using identical set of parameters and augmentations in order to retain fairness. The results are given in Table 5. We again notice a significant margin of improvement over the listed methods and across the batch sizes and epochs. Moreover, we witness a greater robustness to batch size wherein the performance shifts are negligible with varying batch sizes. We further detail additional comparisons against MoCo-v3 A.3.2 and SimCLR A.3.3in the Appendix.

## 5 DISCUSSION

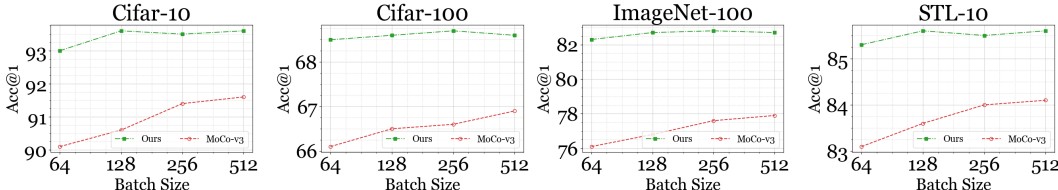

Figure 2: **Batch size robustness.** Linear evaluation results (top-1 accuracy) using various batch sizes on the mentioned datasets compared against MoCo-v3. Our method retains stability in performance with varying batch sizes with relatively smaller variance.

Table 5: Comparison by pretraining on ImageNet and ImageNet-100 against SimCLR (Chen et al., 2020) and SogCLR (Yuan et al., 2022) using the ResNet-50 backbone. Linear evaluation (acc@1) results are listed. We retain the parameter and augmentation settings.

| Batch size | Methods | ImageNet | | | ImageNet-100 | | |
|---|---|---|---|---|---|---|---|
| | Epoch | 100 | 200 | 400 | 100 | 200 | 400 |
| 128 | SimCLR | 62.6 | 64.0 | 64.1 | 68.5 | 72.7 | 75.7 |
| | SogCLR | 64.9 | 66.2 | 67.4 | 72.2 | 76.7 | 79.3 |
| | AUC-CL | **66.1** | **67.6** | **68.8** | **79.0** | **81.4** | **82.7** |
| 256 | SimCLR | 62.8 | 64.3 | 65.7 | 69.7 | 73.6 | 76.1 |
| | SogCLR | 65.2 | 67.1 | 68.7 | 71.8 | 76.3 | 78.7 |
| | AUC-CL | **66.2** | **67.7** | **69.1** | **78.9** | **81.2** | **82.8** |
| 512 | SimCLR | 63.8 | 65.6 | 66.7 | 70.9 | 74.1 | 75.9 |
| | SogCLR | 65.0 | 67.2 | 68.8 | 71.8 | 75.8 | 78.2 |
| | AUC-CL | **66.2** | **67.9** | **69.2** | **78.8** | **81.3** | **82.8** |

**Intuition**  We illustrate a simplified intuition behind our formulation in the Figure 1. The squared terms $A_1$ and $A_2$ enforce the similarity of the positive and negative pairs towards the values of $a$ and $b$. However, we observe that merely minimizing these two components and ignoring $A_3$ leads to a representational collapse. The component $A_3$ therefore serves to counteract this collapse of the objective leading to an increase in the similarities of the positive pairs and a decrease in those of the negatives, thus balancing the effects of each component towards a desirable optimum. A combination of the three terms, leads to representations learned by our framework that would satisfy both positive-data compactness and negative-data repulsion, which is not considered explicitly in standard contrastive learning. Rearranging, the positive terms $\left[(\mathrm{sim}(i,i+) - a)^2 + 2\alpha(1 - \mathrm{sim}(i,i+) \mid y_{ii+})\right]$ encourage the augmented views to be consistent in the feature space whereas $\sum_{j \in \mathcal{B}_i}[(\mathrm{sim}(i,j) - b)^2 + 2\alpha\,\mathrm{sim}(i,j) \mid y_{ij-}]$ forces separation across negatives, encouraging representations to match a high entropy prior distribution.

**Adaptive margin robustness**  The function exhibits robustness to variations in batch size, attributed to unbiased stochastic gradient estimators, as derived theoretically (A.1). Notably, this characteristic isn't exclusive to our formulation. Our function's superior performance is evident, and we attribute this to its inherent nature. The retention in performance (Figure 2) with smaller batch sizes is linked to robustness to increased noise in the batch, where a smaller batch tends to have a noisier set of similarity scores. Notably, our function introduces an adaptive margin term $A_3$, a unique component in our formulation. This term proves particularly relevant for handling noise in stochastic samples. For instance, in cases where the $\mathrm{sim}(i,j)$ term is artificially high due to noise, $\mathrm{sim}(i,i)$ is strategically adjusted to compensate for the noise, improving class clustering in the feature space and mitigating the effects of inherent noise. Additionally, we conduct experiments to establish that the component $A_3$ whose influence is monitored by $\alpha$ is critical to avert a mode collapse in Table 11 in the Appendix.

**Convergence**  We empirically analyze the convergence of our function by comparing against known arts in Figure 3 (Appendix). We train using ResNet-18 on Cifar-10 using the configuration mentioned in A.3.3 and plot the $k$-NN Top-1 accuracy curves for the methods and notice that our method converges noticeably sooner. Specifically, our method outperforms SimSiam (second best) by over 8% at the epoch 100. We conjecture that the component $A_2$ may be instrumental in such rapid early convergence. $A_2$ monitors the uniformity objective Wang & Isola (2020). An unbiased estimate of the component is therefore more asymptotically optimal and thus may be a significant contributing factor in helping the model learn optimal feature spaces earlier than standard methods.

# 6 CONCLUSION

In this work, we investigate the popular SSL via contrastive learning and its variants. We delineate the drawbacks of the corresponding objectives and propose a batchsize-robust objective inspired from AUC maximisation of data. We further analyze our objective for convergence and prove the function's robustness in convergence to batch size. Furthermore, we conduct several experiments under varying settings of representation learning, transfer learning and few shot classification, and compare against the state of the arts to demonstrate the merits of our method agnostic to batch sizes.

## 7 ACKNOWLEDGEMENT

This work is partially supported by NSF AI Institute-2229873, NSF RI-2223292, an Amazon research award, and an Adobe gift fund. Any opinions, findings and conclusions or recommendations expressed in this material are those of the author(s) and do not necessarily reflect the views of the National Science Foundation, the Institute of Education Sciences, or the U.S. Department of Education.

## 8 BROADER IMPACT

The publication on self-supervised contrastive learning introduces a method that exhibits robustness to variations in batch size, advancing the field of unsupervised representation learning. This innovation enhances scalability and efficiency, making deep learning techniques more accessible across diverse computational infrastructures. By democratizing the utilization of self-supervised learning methodologies, the research fosters broader accessibility to cutting-edge technologies and holds potential for significant impacts across various applications, from natural language processing to computer vision.

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

# A APPENDIX

## A.1 PROOF OF 3.3

*Proof.* To optimize the objective in 8 with the standard widely available optimizers, we adopt the following efficient mini-batch stochastic gradient estimator.

$$
\begin{aligned}
\widehat{\nabla}_{\mathbf{w}} L(\mathbf{w}, b; \mathbf{x}_i, \mathcal{A}, \mathcal{A}', \mathcal{B}_i) &= 2\left[(\mathrm{sim}(i, i+) - a)\nabla_{\mathbf{w}}\mathrm{sim}(i, i+) \mid y_{ii+}\right] \\
&+ 2\sum_{j \in \mathcal{B}_i}\left[2(\mathrm{sim}(i, j) - b)\nabla_{\mathbf{w}}\mathrm{sim}(i, j) \mid y_{ij-}\right] + 2\alpha\left[-\nabla_{\mathbf{w}}\mathrm{sim}(i, i+) \mid y_{ii+}\right] \\
&+ 2\alpha\Big[\sum_{j \in \mathcal{B}_i}\nabla_{\mathbf{w}}\mathrm{sim}(i, j) \mid y_{ij-}\Big], \\
\widehat{\nabla}_b L(\mathbf{w}, b; \mathbf{x}_i, \mathcal{A}, \mathcal{A}', \mathcal{B}_i) &= 2\sum_{j \in \mathcal{B}_i}\left[(b - \mathrm{sim}(i, j)) \mid y_{ij-}\right],
\end{aligned}
\tag{9}
$$

where $\mathcal{B}_i$ is a random mini-batch of images drawn from $\mathcal{M}_i$, which still excludes the image $\mathbf{x}_i$ itself. For the analysis, we consider the following simple update

$$
\begin{aligned}
\mathbf{w}_{t+1} &= \mathbf{w}_t - \frac{\beta}{B}\sum_{i \in \mathcal{B}}\widehat{\nabla}_{\mathbf{w}} L(\mathbf{w}, b; \mathbf{x}_i, \mathcal{A}, \mathcal{A}'; \mathcal{B}_i) \\
b_{t+1} &= b_t - \frac{\beta}{B}\sum_{i \in \mathcal{B}}\widehat{\nabla}_b L(\mathbf{w}, b; \mathbf{x}_i, \mathcal{A}, \mathcal{A}', \mathcal{B}_i) .
\end{aligned}
\tag{10}
$$

Our analysis adopts the following assumption. Note that the cosine similarity metric $\mathrm{sim}(i, j)$ for any two augmentations $x_i, x_j$ and two operations $\mathcal{A}, \mathcal{A}'$ is a function of $\mathbf{w}$. Then, for the purpose of analysis, we instead use the notation $\mathrm{sim}(i, j)(\mathbf{w})$ to capture the dependence on $\mathbf{w}$.

**Assumption A.1.** We assume for any $i, j, \mathcal{A}, \mathcal{A}'$, the cosine similarity metric $\mathrm{sim}(i, j)(\mathbf{w})$ satisfies

- $\mathrm{sim}(i, j)(\mathbf{w})$ is $L_0$-Lipschitz continuous and $L_1$-smooth.
- $\max\{a, b\} \leq \tau$,

for some positive constant $\tau > 0$.

Note that the boundedness condition in the second item can be replaced by adding a projection of $\mathbf{w}$ and $b$ onto a bounded set like a ball for the updates in 10. However, for the simplicity, we directly assume the boundedness, which is also observed during the optimization process in the experiments. The global contrastive objective in our case that is based on the entire dataset $\mathcal{D}$ is is given by $\min_{\mathbf{w} \in \mathbf{R}^\mathbf{d}, b}\mathbb{E}_{\mathbf{x}_i \sim \mathcal{D}, \mathcal{A}, \mathcal{A}' \sim \mathcal{P}}[L'_s(\mathbf{w}, b; \mathbf{x}_i, \mathcal{A}, \mathcal{A}', \mathcal{M}_i)]$ where

$$
\begin{aligned}
L(\mathbf{w}, b; \mathbf{x}_i, \mathcal{A}, \mathcal{A}', \mathcal{M}_i) &= \left[(\mathrm{sim}(i, i+) - a)^2 \mid y_{ii+}\right] + \sum_{j \in \mathcal{M}_i}\left[\mathrm{sim}(i, j) - b)^2 \mid y_{ij-}\right] \\
&+ \left\{2\alpha\left[1 - \mathrm{sim}(i, i+) \mid y_{ii+} + \sum_{j \in \mathcal{M}_i}\mathrm{sim}(i, j) \mid y_{ij-}\right] - \alpha^2\right\}.
\end{aligned}
$$

We first prove the smoothness of this objective function. Note that the cosine similarity $\mathrm{sim}(i, j)(\mathbf{w})$ is bounded by 1, and hence we have $\max\{|\mathrm{sim}(i, j)(\mathbf{w})|, a, b\} \leq \tau + 1$. Then, the gradient of the this objective takes the form of

$$
\begin{aligned}
\nabla_{\mathbf{w}} L(\mathbf{w}, b; \mathbf{x}_i, \mathcal{A}, \mathcal{A}', \mathcal{M}_i) &= \left[2(\mathrm{sim}(i, i+) - a)\nabla_{\mathbf{w}}\mathrm{sim}(i, i+) \mid y_{ii+}\right] \\
&+ \sum_{j \in \mathcal{M}_i}\left[2(\mathrm{sim}(i, j) - b)\nabla_{\mathbf{w}}\mathrm{sim}(i, j) \mid y_{ij-}\right] \\
&+ 2\alpha\left[-\nabla_{\mathbf{w}}\mathrm{sim}(i, i+) \mid y_{ii+}\right] + 2\alpha\Big[\sum_{j \in \mathcal{M}_i}\nabla_{\mathbf{w}}\mathrm{sim}(i, j) \mid y_{ij-}\Big], \\
\nabla_b L(\mathbf{w}, b; \mathbf{x}_i, \mathcal{A}, \mathcal{A}', \mathcal{M}_i) &= 2\sum_{j \in \mathcal{M}_i}\left[(b - \mathrm{sim}(i, j)) \mid y_{ij-}\right].
\end{aligned}
$$

Based on the gradient form here, we can obtain, for any two parameters $(\mathbf{w}, b), (\mathbf{w}', b')$

$$
\begin{aligned}
&\|\nabla_{\mathbf{w}} L(\mathbf{w}, b; \mathbf{x}_i, \mathcal{A}, \mathcal{A}', \mathcal{M}_i) - \nabla_{\mathbf{w}} L(\mathbf{w}', b'; \mathbf{x}_i, \mathcal{A}, \mathcal{A}', \mathcal{M}_i)\| \\
&\leq \big\| \left[ 2(\text{sim}(i, i+)(\mathbf{w}) - a) \nabla_{\mathbf{w}} \text{sim}(i, i+)(\mathbf{w}) \mid y_{ii+} \right] - \\
&\quad \left[ 2(\text{sim}(i, i+)(\mathbf{w}') - a) \nabla_{\mathbf{w}} \text{sim}(i, i+)(\mathbf{w}') \mid y_{ii+} \right] \big\| \\
&\quad + \sum_{j \in \mathcal{M}_i} \big\| \left[ 2(\text{sim}(i, j)(\mathbf{w}) - b) \nabla_{\mathbf{w}} \text{sim}(i, j)(\mathbf{w}) \mid y_{ij-} \right] - \\
&\quad \left[ 2(\text{sim}(i, j)(\mathbf{w}') - b') \nabla_{\mathbf{w}} \text{sim}(i, j)(\mathbf{w}') \mid y_{ij-} \right] \big\| \\
&\quad + 2\alpha \big\| \left[ -\nabla_{\mathbf{w}} \text{sim}(i, i+)(\mathbf{w}) \mid y_{ii+} \right] - \left[ -\nabla_{\mathbf{w}} \text{sim}(i, i+)(\mathbf{w}') \mid y_{ii+} \right] \big\| \\
&\quad + 2\alpha \sum_{j \in \mathcal{M}_i} \big\| \left[ \nabla_{\mathbf{w}} \text{sim}(i, j)(\mathbf{w}) \mid y_{ij-} \right] - \left[ \nabla_{\mathbf{w}} \text{sim}(i, j)(\mathbf{w}') \mid y_{ij-} \right] \big\|,
\end{aligned}
$$

which, in conjunction with A.1 and using the fact that $ab - a'b' = a(b - b') + (a - a')b'$, yields

$$
\begin{aligned}
&\|\nabla_{\mathbf{w}} L(\mathbf{w}, b; \mathbf{x}_i, \mathcal{A}, \mathcal{A}', \mathcal{M}_i) - \nabla_{\mathbf{w}} L(\mathbf{w}', b; \mathbf{x}_i, \mathcal{A}, \mathcal{A}'), \mathcal{M}_i\| \\
&\leq 4(\tau + 1)L_1 n \|\mathbf{w} - \mathbf{w}'\| + 2L_0^2 n \|\mathbf{w} - \mathbf{w}'\| + 2(n-1)L_0 |b - b'| + 2\alpha n L_1 \|\mathbf{w} - \mathbf{w}'\| \\
&\leq (4(\tau + 1)L_1 n + 2L_0^2 n + 2\alpha n L_1)\|\mathbf{w} - \mathbf{w}'\| + 2(n-1)L_0 |b - b'| \\
&\leq \underbrace{\sqrt{2}(4(\tau + 1)L_1 n + 2L_0(L_0 + 1)n + 2\alpha n L_1)}_{L_w} \sqrt{\|\mathbf{w} - \mathbf{w}'\|^2 + |b - b'|^2}
\end{aligned}
\tag{11}
$$

Similarly, for the gradient w.r.t. $b$, we have

$$
\begin{aligned}
&\|\nabla_b L(\mathbf{w}, b; \mathbf{x}_i, \mathcal{A}, \mathcal{A}', \mathcal{M}_i) - \nabla_b L(\mathbf{w}', b'; \mathbf{x}_i, \mathcal{A}, \mathcal{A}', \mathcal{M}_i)\| \\
&\qquad\qquad\qquad \leq 2(n-1)|b - b'| + 2(n-1)L_0 \|\mathbf{w} - \mathbf{w}'\| \\
&\qquad\qquad\qquad \leq \underbrace{2n(L_0 + 1)\sqrt{2}}_{L_b} \sqrt{\|\mathbf{w} - \mathbf{w}'\|^2 + |b - b'|^2}
\end{aligned}
\tag{12}
$$

First note that our stochastic gradient estimator $\widehat{\nabla}_{\mathbf{w}} L(\mathbf{w}, b; \mathbf{x}_i, \mathcal{A}, \mathcal{A}', \mathcal{B}_i)$ and $\widehat{\nabla}_b L(\mathbf{w}, b; \mathbf{x}_i, \mathcal{A}, \mathcal{A}', \mathcal{B}_i)$ are unbiased estimators. To see this, based on the forms in 9, we have

$$
\begin{aligned}
\mathbb{E}\widehat{\nabla}_{\mathbf{w}} L(\mathbf{w}, b; \mathbf{x}_i, \mathcal{A}, \mathcal{A}', \mathcal{B}_i) &= \mathbb{E}[\mathbb{E}\widehat{\nabla}_{\mathbf{w}} L(\mathbf{w}, b; \mathbf{x}_i, \mathcal{A}, \mathcal{A}', \mathcal{B}_i) \mid x_i, \mathcal{A}, \mathcal{A}'] \\
&= \mathbb{E}[\mathbb{E}\nabla_{\mathbf{w}} L(\mathbf{w}, b; \mathbf{x}_i, \mathcal{A}, \mathcal{A}') \mid x_i, \mathcal{A}, \mathcal{A}'] \\
&= \nabla_{\mathbf{w}} L(\mathbf{w}, b)
\end{aligned}
$$

where the first equality follows because $\mathcal{B}_i$ is sampled from $\mathcal{M}_i$. A similar result is obtained for $\widehat{\nabla}_b L(\mathbf{w}, b; \mathbf{x}_i, \mathcal{A}, \mathcal{A}', \mathcal{B}_i)$, i.e., $\mathbb{E}\widehat{\nabla}_b L(\mathbf{w}, b; \mathbf{x}_i, \mathcal{A}, \mathcal{A}', \mathcal{B}_i) = \nabla_b L(\mathbf{w}, b)$.

Based on the smoothness results in 11 and 12 and the unbiased estimation, we are now ready to prove the main theorem. Let $\mathbf{v} = (\mathbf{w}, b)$ denote all optimization parameters. From the smoothness results in 11 and 12, we can establish the smoothness of the overall objective $L(\mathbf{v}) = L(\mathbf{w}, b)$ as below. For any $\mathbf{v}$ and $\mathbf{v}'$,

$$
\|\nabla L(\mathbf{v}) - \nabla L(\mathbf{v}')\| \leq \sqrt{L_b^2 + L_w^2} \|\mathbf{v} - \mathbf{v}'\|.
\tag{13}
$$

Then, based on 13, we have

$$
L(\mathbf{v}_{t+1}) \leq L(\mathbf{v}_t) + \langle \mathbf{v}_{t+1} - \mathbf{v}_t, \nabla L(\mathbf{v}_t) \rangle + \frac{\sqrt{L_b^2 + L_w^2}}{2} \|\mathbf{v}_{t+1} - \mathbf{v}_t\|^2,
$$

which, by taking the expectation $\mathbb{E}_t := \mathbb{E}[\cdot|v_t]$ on the both sides and using our unbiased gradient estimators, yields

$$
\begin{aligned}
\mathbb{E}_t L(\mathbf{v}_{t+1}) \leq & L(\mathbf{v}_t) - \beta\|\nabla L(\mathbf{v}_t)\|^2 + \frac{\sqrt{L_b^2 + L_w^2}}{2}\mathbb{E}_t\|\mathbf{v}_{t+1} - \mathbf{v}_t\|^2 \\
= & L(\mathbf{v}_t) - \beta\|\nabla L(\mathbf{v}_t)\|^2 + \frac{\sqrt{L_b^2 + L_w^2}}{2}\beta^2\mathbb{E}_t\Big(\Big\|\frac{1}{B}\sum_{i\in\mathcal{B}}\widehat{\nabla}_b L(\mathbf{w}, b; \mathbf{x}_i, \mathcal{A}, \mathcal{A}', \mathcal{B}_i)\Big\|^2 \\
& + \Big\|\frac{1}{B}\sum_{i\in\mathcal{B}}\widehat{\nabla}_\mathbf{w} L(\mathbf{w}, b; \mathbf{x}_i, \mathcal{A}, \mathcal{A}'; \mathcal{B}_i)\Big\|^2\Big).
\end{aligned}
\tag{14}
$$

Note from the forms in 9 and A.1, we have

$$
\Big\|\frac{1}{B}\sum_{i\in\mathcal{B}}\widehat{\nabla}_\mathbf{w} L(\mathbf{w}, b; \mathbf{x}_i, \mathcal{A}, \mathcal{A}'; \mathcal{B}_i)\Big\|^2 \leq \Big\|\widehat{\nabla}_\mathbf{w} L(\mathbf{w}, b; \mathbf{x}_i, \mathcal{A}, \mathcal{A}'; \mathcal{B}_i)\Big\|^2
$$

$$
\leq 8(\tau+1)nL_0 + 2\alpha nL_0
$$

$$
\Big\|\frac{1}{B}\sum_{i\in\mathcal{B}}\widehat{\nabla}_b L(\mathbf{w}, b; \mathbf{x}_i, \mathcal{A}, \mathcal{A}'; \mathcal{B}_i)\Big\|^2 \leq \Big\|\widehat{\nabla}_b L(\mathbf{w}, b; \mathbf{x}_i, \mathcal{A}, \mathcal{A}'; \mathcal{B}_i)\Big\|^2 \leq 4n(\tau+1).
\tag{15}
$$

Incorporating 15 into 14 yields

$$
\mathbb{E}_t L(\mathbf{v}_{t+1}) \leq L(\mathbf{v}_t) - \beta\|\nabla L(\mathbf{v}_t)\|^2 + \beta^2\frac{\sqrt{L_b^2 + L_w^2}}{2}(4n(\tau+1) + 8(\tau+1)nL_0 + 2\alpha nL_0).
$$

Unconditioning on $\mathbf{v}_t$, rearranging the above inequality and doing the telescoping over $t$ from $0$ to $T-1$, we have

$$
\frac{1}{T}\sum_{t=0}^{T-1}\|\nabla L(\mathbf{v}_t)\|^2 \leq \frac{L(\mathbf{v}_0) - \min_\mathbf{v} L(\mathbf{v})}{\beta T} + \beta\frac{\sqrt{L_b^2 + L_w^2}}{2}(4n(\tau+1) + 8(\tau+1)nL_0 + 2\alpha nL_0)
$$

$$
\leq \mathcal{O}(\frac{1}{\beta T} + \beta)
\tag{16}
$$

which, in conjunction with $\beta = \frac{1}{\sqrt{T}}$ and the definition of $t'$, finishes the proof. $\qquad\square$

## A.2 AUC OPTIMIZATION AND CONTRASTIVE LEARNING

We further elucidate our motivations behind adaptation of the AUC optimization framework towards contrastive learning. AUC as a metric was formulated for binary classification wherein, the objective of the network is to enhance the prediction scores for "positive" samples in comparison to the "negatives" (Equation 3.1). Thus by virtue of its construction, it aligns seamlessly for an application in contrastive learning wherein due to the lack of labels, one is compelled to enforce separation amongst classes through a binary objective with "positives" being the augmentations of the same sample and "negatives", the augmentations of other samples within the batch. Additionally, AUC was originally devised to address the imbalance of classes whereby accuracy as a metric may lead to misleading evaluation of the network. A classic example of this phenomenon is often cited with a dataset containing 100 samples, 99 of which are of the "positive" class and a network that predicts every sample as a "positive" will therefore have attained a 99% accuracy. This aspect of the function resonates well with the context of contrastive learning in our application, as for one image in our batch of samples, the remaining images are considered to be "negative".

## A.3 ADDITIONAL RESULTS

### A.3.1 LONGER PRETRAINING

In order to attain stronger convergence, we conduct the pretraining procedure using our method for 800 epochs on ImageNet with the ResNet-50 backbone. In Table 6, we compare against the prominent methods for SSL. We retain the batch size of 256 for our method, wherein the remaining methods have been trained using a larger batch size of 1024. The parameters and setup of the method follows the description in 4. Yet again, our loss function avails a superior result using a far smaller batch size.

Table 6: **ImageNet Longer Pretraining (ResNet-50)**. Top-1 accuracy for linear evaluation results are listed. We conduct pretraining for 800 epochs and compare against known arts.

| Method | Batch Size | 800 ep |
|---|---|---|
| SimCLR Chen et al. (2020) | 1024 | 69.1 |
| InfoMin Poole et al. (2020) | 1024 | 73.0 |
| BarlowTwins Zbontar et al. (2021) | 1024 | 73.2 |
| OBOW Gidaris et al. (2020) | 1024 | 73.8 |
| BYOL Grill et al. (2020) | 1024 | 74.4 |
| DCv2 Caron et al. (2018) | 1024 | 75.2 |
| SwAV Caron et al. (2020) | 1024 | 75.3 |
| DINO Caron et al. (2021) | 1024 | 75.3 |
| AUC-CL | 256 | **75.5** |

### A.3.2 COMPARISON AGAINST MOCO-V3

**Pretraining**   We train the MoCo-v3 architecture by replacing the objective function with ours whilst retaining the architecture and parameter settings. Our results are based on a batch size of 128 on the datasets Cifar-10, ImageNet-S and ImageNet using the ViT-small backbone, which are illustrated in Table 7. Our models were trained for 100 epochs with the default parameter and augmentation settings. Our model consistently outperforms the results when comparing to MoCo-v3 by values higher than 2% for Cifar-10 and Cifar-100 datasets and by over 5% for ImageNet.

Table 7: **Pre-training and linear evaluation vs MoCo-v3.** 'a/b/c' in the kNN column are acc. (%) with k = 10, 20, 100, respectively. The Cifar results use the ResNet-18 backbone and the ImageNet results use the Vit-Small backbone.

| Dataset | MoCo-v3 | | AUC-CL | |
|---|---|---|---|---|
| | KNN | Linear | KNN | Linear |
| Cifar-10 | 84.6/84.5/84.4 | 91.4 | 87.6/87.2/86.1 | 93.6 |
| Cifar-100 | 51.3/53.5/52.7 | 66.6 | 52.9/54.7/53.4 | 69.7 |
| ImageNet-100 | 74.1/74.7/73.8 | 77.6 | 77.0/77.6/76.9 | 82.5 |
| ImageNet | 50.0/50.6/49.1 | 62.3 | 54.1/54.0/51.4 | 67.9 |

**Transfer Learning**   We subsequently evaluate the model for transfer learning on the Cifar-10, Cifar-100 datasets, Flowers-102 Nilsback & Zisserman (2008) and the Pets Parkhi et al. (2012) datasets after pretraining on ImageNet. The protocol followed is identical as mentioned in Chen et al. (2021) and the results are listed under Table 8

Table 8: **Transfer Learning comparison with MoCov3** after pretraining on ImageNet. Numbers next to the method indicate the batch size used. 'a/b' represent the Top-1 and Top-5 accuracies (%). The numbers listed under the 'Supervised' category are borrowed from Dosovitskiy et al. (2020).

| | Cifar-10 | Cifar-100 | Flowers-102 | Pets | Average |
|---|---|---|---|---|---|
| Random Init. | 77.8 | 48.5 | 54.4 | 40.1 | 55.2 |
| Supervised | 98.1 | 87.1 | 89.5 | 93.8 | 92.1 |
| Moco-v3-256 | 97.1/100.0 | 84.6/97.7 | 88.6/96.7 | 85.0/98.6 | 88.8/98.2 |
| AUC-CL-128 | 98.2/100.0 | 85.4/97.9 | 88.9/97.5 | 87.1/99.3 | 89.9/98.7 |

### A.3.3 COMPARISON AGAINST SIMCLR

We illustrate the performance of our method on Cifar-10, Cifar-100 and STL-10 in Table 9, comparing against SimCLR for varying batch sizes and epochs. In this experiment, we use a backbone of ResNet-50. As is standard practice, we replace the first $7 \times 7$ Conv layer with stride 2 with $3 \times 3$ with stride 1 and remove the first max-pooling layer. The learning rate is fixed to $1e - 3$. The training is conducted

Table 9: **SimCLR comparison.** Top-1 KNN evaluation results on Cifar-10, Cifar-100 and STL-10 datasets.

| Dataset | BS Epoch | 64 | | | 128 | | | 256 | | |
|---|---|---|---|---|---|---|---|---|---|---|
| | | 200 | 300 | 500 | 200 | 300 | 500 | 200 | 300 | 500 |
| Cifar-10 | SimCLR | 79.6 | 81.8 | 84.15 | 82.8 | 83.4 | 86.2 | 83.8 | 85.8 | 86.9 |
| | AUC-CL | 85.4 | 87.4 | 89.1 | 86.1 | 87.9 | 89.4 | 85.7 | 87.3 | 88.7 |
| Cifar-100 | SimCLR | 45.2 | 47.8 | 49.9 | 47.8 | 53.5 | 55.6 | 48.5 | 54.1 | 56.0 |
| | AUC-CL | 52.9 | 56.0 | 57.2 | 53.7 | 56.1 | 57.3 | 53.7 | 55.9 | 57.1 |
| STL-10 | SimCLR | 69.8 | 69.7 | 73.6 | 72.1 | 72.9 | 74.1 | 75.6 | 75.7 | 76.2 |
| | AUC-CL | 76.1 | 76.2 | 78.6 | 77.5 | 78.9 | 80.9 | 76.8 | 78.4 | 80.4 |

Table 10: **Comparison against DCL** for the listed batch sizes and datasets. The linear evaluation scores following the protocol mentioned in DCL are listed. All models are trained for 200 epochs. The backbone used for ImageNet was ResNet-50 whereas for the other datasets, the backbone was set to ResNet-18.

| Method | BS/Dataset | 128 | 256 | 512 |
|---|---|---|---|---|
| DCL | ImageNet | 64.3 | 65.9 | 65.8 |
| AUC-CL | ImageNet | 67.6 | 67.7 | 67.9 |
| DCL | Cifar-10 | 85.7 | 85.3 | 84.7 |
| AUC-CL | Cifar-10 | 88.4 | 87.9 | 88.3 |
| DCL | Cifar-100 | 58.9 | 58.5 | 58.4 |
| AUC-CL | Cifar-100 | 59.3 | 59.5 | 59.5 |
| DCL | STL-10 | 86.1 | 85.7 | 85.6 |
| AUC-CL | STL-10 | 86.5 | 86.3 | 86.5 |

for 500 epochs. We notice that our method significantly outperforms SimCLR for smaller batch sizes by margins of $+7\%$ for both datasets, and retains a consistent performance across the batch sizes. Upon convergence, our method outperforms SimCLR in Top-1 accuracy by an average of $3.8\%$. Subsequent to the epoch 20, our method overcomes the performance of SimCLR, and towards the end of training outperforms by over $3.2\%$. This trend is reflected across the datasets and batch-size, epoch settings, with the margins of out-performance particularly stark for smaller batch sizes.

### A.3.4 COMPARISON AGAINST DCL

We compare against the popular objective of DCL Yeh et al. (2022) directly for the datasets of ImageNet, Cifar and STL. Here, identical to their work, we pretrain ResNet-50 and ResNet-18 architectures for ImageNet and the rest respectively, for 200 epochs. We follow the augmentation parameters as per their work in order to retain fairness and train our models for varying batch sizes. The results are listed in Table 10. We again witness a substantial margin of improvement over DCL across the comparisons.

### A.3.5 ROBUSTNESS TO $\alpha$

We conduct extensive ablations to experiment with the significance of the component $A_3$ in our main formulation 8. We train the model using the parameters and architecture described in section A.3.3 with a batch size of 128 on Cifar-10, while varying the value of the parameter $\alpha$ which modulates the influence of the component. The results are illustrated in Table 11. Here we show the KNN evaluation results for at several epochs during training retaining identical settings. We establish that $A_3$ is crucial to our formulation and prevents the mode collapse phenomenon often observed in SSL frameworks, where the features are mapped to a unique point in the hypersphere regardless of the class distinction. Moreover, for all other values of $\alpha$, our formulation retains its performance across

Table 11: **Robustness to** $\alpha$. We evaluate our loss function on the Cifar-10 dataset for varying values of $\alpha$ using the KNN protocol for k=200. The values are the Top-1 KNN evaluation accuracies at various epochs and $\alpha$ parameter settings.

| Epoch/$\alpha$ | 0.0 | 0.1 | 0.5 | 0.7 | 1.0 |
|---|---|---|---|---|---|
| 50 | 25.1 | 77.7 | 78.3 | 77.6 | 76.8 |
| 100 | 10.0 | 82.9 | 83.4 | 82.4 | 81.5 |
| 150 | 10.00 | 85.18 | 85.70 | 84.76 | 84.41 |
| 200 | 10.00 | 86.80 | 86.61 | 85.89 | 85.98 |
| 250 | 10.00 | 87.40 | 87.64 | 87.21 | 87.06 |

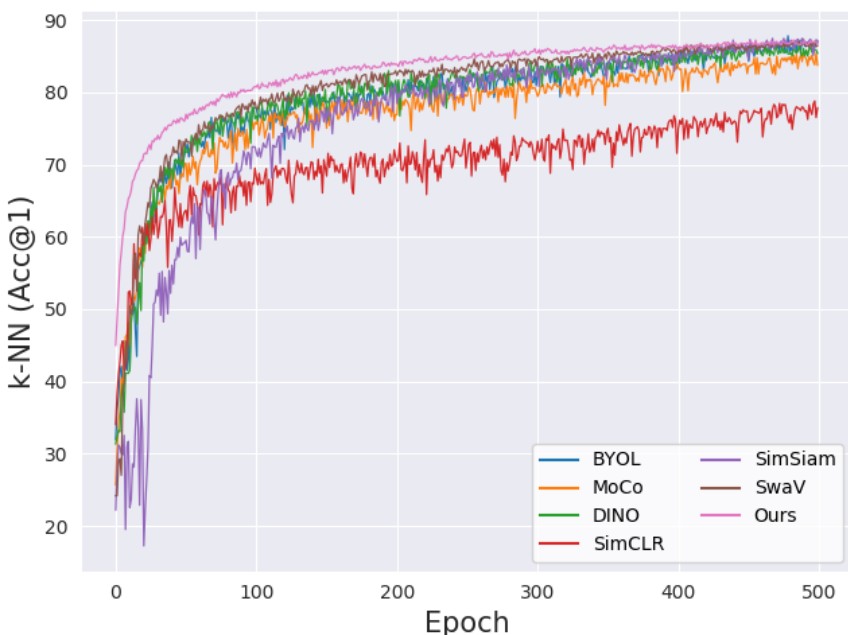

Figure 3: **$k$-NN curves:** Plot of the $k$-NN accuracy curves for various methods trained on Cifar-10 for 500 epochs using ResNet-18. The values for the other methods were borrowed from LightlyAI

the epochs, with nominal differences, thus illustrating that the component is crucial to the formulation as well as robust to variations in $\alpha$, which therefore requires no additional tuning.

