# OpenReview forum: "AUC-CL: A Batchsize-Robust Framework for Self-Supervised Contrastive Representation Learning"
_ICLR.cc/2024/Conference — ICLR 2024 poster_

### Official Review · Reviewer_QVtp · 2023-10-23

**Soundness:** 3 good
**Presentation:** 2 fair
**Contribution:** 2 fair
**Rating:** 6
**Confidence:** 4

**Summary:**

This paper proposes AUC-CL, a new batch-size robust framework for self-supervised contrastive representation learning. At first, the authors point out the limitations of existing NT-Xent objectives (i.e., performance is heavily influenced by batch size). Then, the authors theoretically analyze why SimCLR loss is sensitive to batch size from the gradient perspective. Finally, the author provides the theoretical analysis to demonstrate the convergence guarantee of AUC-CL. Although the relative performance is significant, the absolute accuracy is not competitive. Besides, the reasons why AUC-CL is batch-size robust are not clear, which seems to be an important property.

**Strengths:**

The paper is well-organized and easy to follow

The experimental results demonstrate the effectiveness of the proposed AUC-CL.

**Weaknesses:**

1. As stated in the summary, the motivations behind using the AUC framework in combination with contrastive learning, and why this is helpful for batch size robustness, are not clear.

**Questions:**

As I have reviewed this submission in the other venue, and most of the concerns have been addressed, I have no further questions.

---

> ### Author Response · Authors · 2023-11-17
>
> **Although the relative performance is significant, the absolute accuracy is not competitive.**
>
> We express our appreciation to the reviewer for their favorable review and comments. We direct the reviewer to our primary response, where the ongoing training of ResNet-50 on ImageNet is elaborated. Furthermore, we earnestly request the reviewer to extend their evaluation beyond larger architectures and batch sizes, emphasizing the distinctive value of our contribution to the field of contrastive learning literature.
>
> **The motivations behind using the AUC framework in combination with contrastive learning, and why this is helpful for batch size robustness, are not clear.**
>
> We thank the reviewer for this critique and clarify further. The AUC framework fits perfectly for our application for the following reasons.
>
> 1. AUC is formulated for binary classification problems wherein, the objective of the network is to enhance the prediction scores for “positive” samples in comparison to the “negatives” (Definition 3.1 in manuscript). Thus by virtue of its construction, it aligns seamlessly for our application in contrastive learning wherein due to the lack of labels, we are compelled to enforce separation amongst classes through a binary objective (positives=augmentations of the same sample, negatives=augmentations of different samples).
>
> 2. AUC was originally devised to address the imbalance of classes whereby accuracy as a metric may lead to misleading evaluation of the network. A classic example of this phenomenon is often cited with a dataset containing 100 samples, 99 of which are of the “positive” class and a network that predicts every sample as a “positive” will therefore have attained a 99\% accuracy. This aspect of the function resonates well with our context as for one image in our batch of samples, the remaining images are considered to be “negative”.
>
>
> The function exhibits robustness to variations in batch size, attributed to unbiased stochastic estimators providing gradient estimates that remain unbiased for the entire batch, as theoretically demonstrated in our manuscript (section A.1). Notably, this characteristic isn't exclusive to our formulation, as other works, such as [1, 2, 3, 4, 5, 6], also integrate unbiased estimators in their loss functions. Our function's superior performance is evident, and we attribute this to its inherent nature. The decline in performance observed in other formulations with smaller batch sizes is linked to increased noise in the batch, where a smaller batch tends to have a noisier set of similarity scores. Notably, our function introduces an adaptive margin term ($A_3$), a unique component in our formulation. This term proves particularly relevant for handling noise in stochastic samples. For instance, in cases where the sim$(i, j)$ term is artificially high due to noise, the term sim$(i, i)$ is strategically adjusted to compensate for the noise, improving class clustering in the feature space and mitigating the effects of inherent noise.
>
> [1] Grill, Jean-Bastien, et al. "Bootstrap your own latent-a new approach to self-supervised learning." Advances in neural information processing systems 33 (2020): 21271-21284.
>
> [2] Zbontar, Jure, et al. "Barlow twins: Self-supervised learning via redundancy reduction." International Conference on Machine Learning. PMLR, 2021.
>
> [3] Caron, Mathilde, et al. "Emerging properties in self-supervised vision transformers." Proceedings of the IEEE/CVF international conference on computer vision. 2021.
>
> [4] Zhang, Shaofeng, et al. "Zero-cl: Instance and feature decorrelation for negative-free symmetric contrastive learning." International Conference on Learning Representations. 2021.
>
> [5] Ermolov, Aleksandr, et al. "Whitening for self-supervised representation learning." International Conference on Machine Learning. PMLR, 2021.
>
> [6] Zhang, Shaofeng, et al. "Align representations with base: A new approach to self-supervised learning." Proceedings of the IEEE/CVF Conference on Computer Vision and Pattern Recognition. 2022.

---

> > ### Comment · Reviewer_QVtp · 2023-11-18
> > **Thanks for the response.**
> >
> > I appreciate the authors' detailed responses, which clarify the advantages and motivation of the proposed AUC-CL. Therefore, I would like to change my score to 6.

---

> > > ### Author Response · Authors · 2023-11-18
> > > **Thanks so much for your feedback**
> > >
> > > Dear Reviewer QVtp,
> > >
> > > Thanks so much for reading our response and for raising your score. We will take your comments and suggestions into the revision.
> > >
> > > Best,
> > > Authors

---

### Official Review · Reviewer_pifE · 2023-10-27

**Soundness:** 3 good
**Presentation:** 3 good
**Contribution:** 2 fair
**Rating:** 6
**Confidence:** 4

**Summary:**

This paper proposes AUC-Contrastive Learning, which incorporates the contrastive objective within the AUC-maximization framework. Since it maintains unbiased stochastic gradients, it is more robust to batch sizes compared to the standard contrastive loss. It empirically shows that the method with a batch size of 256 outperforms or is on par with several state-of-the-art methods using larger batch sizes.

**Strengths:**

- The method is novel, and theoretically sound. It is great that there is clear link between the AUC-maximization and robustness to batch size.

- The empirical results are promising. The proposed method with a small batch size can outperform or is on par with state-of-the-art methods with large batch size, which is beneficial for the efficiency of computation power.

**Weaknesses:**

- Some baselines are missing in the experiments. There are other methods for self-supervised representation learning, such as MAE[1] and MAGE[2]. Some of them have better performance than the contrastive learning methods. Although this paper focuses on improving contrastive representation learning, it is still necessary to compare with non-contrastive representation learning methods.

- Some contrastive learning methods which do not need negative samples, e.g. BYOL and DINO, also shows some robustness to batch size. I think it is necessary to list their performance with smaller batch size (e.g., 256, the same as the proposed method) in Table 1.

- Why does the paper use ViT-S on ImageNet in the experiment? As far as I aware, most of the representation learning methods use ViT-B or ViT-L for ImageNet evaluation. Also, do all methods converge at 400 epochs for ResNet-50 and 300 epochs for ViT-S? If the baseline methods can reach the same performance as the proposed method with longer epochs and larger architecture, then I don't think the paper can claim the outperformance over them.

[1] He et al. Masked Autoencoders Are Scalable Vision Learners. CVPR 2022.

[2] Li et al. MAGE: MAsked Generative Encoder to Unify Representation Learning and Image Synthesis. CVPR 2023.

**Questions:**

See Weaknesses.

---

> ### Author Response · Authors · 2023-11-17
>
> **Some baselines are missing in the experiments. There are other methods for self-supervised representation learning, such as MAE and MAGE. Some of them have better performance than the contrastive learning methods. Although this paper focuses on improving contrastive representation learning, it is still necessary to compare with non-contrastive representation learning methods.**
>
> We express appreciation to the reviewer for their valuable feedback and recommendations. We would like to direct the reviewer's attention to the comprehensive comparison in our manuscript, where we assess non-contrastive methods such as BYOL [1], Barlow Twins [2], DINO [3], Zero-CL [4], W-MSE [5], and ARB [6] across various architectures and datasets. Given the common practice of employing ResNet-50 in Self-Supervised Learning (SSL), our comparisons are specifically tailored to include well-known methods using this backbone. Notably, the methods MAE and MAGE, which utilize ViT-H and ViT-L backbones with prohibitively high batch sizes (2048-4096), deviate from this common practice and necessitate industrial-grade equipment for efficient execution. We kindly hope that the reviewer considers the detailed comparisons in Tables 1, 2, and 3 in our manuscript, as these directly address the raised concern and provide insights into our method's performance against recent state-of-the-art techniques.
>
> **Why does the paper use ViT-S on ImageNet in the experiment? As far as I aware, most of the representation learning methods use ViT-B or ViT-L for ImageNet evaluation.**
>
> We respectfully disagree with the reviewer as a substantial number of self-supervised learning methods predominantly utilize ResNet-50 on the ImageNet dataset. Notable recent methods, referenced in [1, 2, 3, 4, 5, 6], form a part of our comparisons in Table 1. A more contemporary trend in Self-Supervised Learning (SSL) involves incorporating the Vision Transformer (ViT), exemplified in DINO (Table 2). It is noteworthy that this comparison only includes a subset of methods compared to those utilizing ResNet-50, signifying a recency in the adoption of the ViT architecture.
> Additionally we would like to mention that incorporating the ViT-B and ViT-L architectures mandate large GPUs for training in a reasonable amount of time which is an infeasible task for us considering our constraints on resources.
>
> **Some contrastive learning methods which do not need negative samples, e.g. BYOL and DINO, also shows some robustness to batch size. I think it is necessary to list their performance with smaller batch size (e.g., 256, the same as the proposed method) in Table 1.**
>
> We appreciate the reviewer's suggestion in this regard. The following are some of the linear evaluation results upon pretraining for 100 epochs on ImageNet using ResNet-50. We also include our results for 100 epoch pretraining here.
>
> |Method	|Batch Size	|Linear Acc@1|
> |:----|:----:|:----:|
> |DINO	|256		|68.2|
> |BYOL	|256		|62.4|
> |VicReg	|256		|63.0|
> |Barlow Twins	|256	|62.9|
> |Ours |256 |**69.5**|
>
> Kindly note that the numbers are borrowed from the popular benchmark of https://docs.lightly.ai/self-supervised-learning/getting_started/benchmarks.html#imagenet in the interest of time. We shall duly incorporate these in our Table 1.
>
>
> [1] Grill, Jean-Bastien, et al. "Bootstrap your own latent-a new approach to self-supervised learning." Advances in neural information processing systems 33 (2020): 21271-21284.
>
> [2] Zbontar, Jure, et al. "Barlow twins: Self-supervised learning via redundancy reduction." International Conference on Machine Learning. PMLR, 2021.
>
> [3] Caron, Mathilde, et al. "Emerging properties in self-supervised vision transformers." Proceedings of the IEEE/CVF international conference on computer vision. 2021.
>
> [4] Zhang, Shaofeng, et al. "Zero-cl: Instance and feature decorrelation for negative-free symmetric contrastive learning." International Conference on Learning Representations. 2021.
>
> [5]  Ermolov, Aleksandr, et al. "Whitening for self-supervised representation learning." International Conference on Machine Learning. PMLR, 2021.
>
> [6] Zhang, Shaofeng, et al. "Align representations with base: A new approach to self-supervised learning." Proceedings of the IEEE/CVF Conference on Computer Vision and Pattern Recognition. 2022.

---

> ### Author Response · Authors · 2023-11-17
>
> **Do all methods converge at 400 epochs for ResNet-50 and 300 epochs for ViT-S? If the baseline methods can reach the same performance as the proposed method with longer epochs and larger architecture, then I don't think the paper can claim the outperformance over them.**
>
> To enhance convergence, our method is trained for 650 epochs using the ResNet-50 architecture on ImageNet, with detailed results outlined in the common response section. We only claim state of the art performance for the widely considered architectures of ResNet-50, ResNet-18 (smaller datasets) and ViT-S, in keeping with the common convention. Additionally, It is crucial to emphasize that our contribution is not geared towards surpassing State-of-the-Art (SOTA) performances but rather proposing a novel contrastive learning approach that significantly reduces computational requirements. We kindly hope that the reviewer acknowledges this distinction and considers our method's unique merits.

---

> ### Author Response · Authors · 2023-11-22
> **Gentle reminder**
>
> Dear reviewer pifE,
>
> We express our gratitude for your dedicated efforts in reviewing and providing valuable suggestions. Despite the prolonged discussion period, we have yet to receive feedback on your response. Kindly note that we have concluded the pretraining of Resnet-50, as outlined in the main response, and have incorporated the necessary updates into the PDF (appendix, Table 6), including minor formatting adjustments. We would really appreciate it if you can let us know if our responses and the longer pretraining results resolve your concerns. Additionally, please inform us of any lingering concerns you may have.
>
> Best regards, Authors

---

> > ### Comment · Reviewer_pifE · 2023-11-22
> > **Thanks for your response**
> >
> > The authors' responses address most of my concerns, while I'd like to clarify that for the ViT-S question I was referring to that most of the representation learning methods use ViT-B or ViT-L for ImageNet evaluation with ViT backbones (besides ResNet backbones). I still think including larger architectures and longer training epochs will enhance the evaluation of this paper. I acknowledge the results with longer epochs in the common response and I think that should be a crucial part for the evaluation.
> >
> > Nevertheless, I appreciate the novelty of the proposed method and understand the limitation of computation resources, so I keep my rating as positive.

---

> > > ### Author Response · Authors · 2023-11-22
> > > **Thanks so much for your feedback**
> > >
> > > Dear Reviewer pifE,
> > >
> > > Thanks so much for your further clarification and for your understanding!  We agree with your point and will definitely incorporate your comments and suggestions into our revision!
> > >
> > > Best,
> > > Authors

---

### Official Review · Reviewer_nw59 · 2023-10-29

**Soundness:** 2 fair
**Presentation:** 2 fair
**Contribution:** 3 good
**Rating:** 5
**Confidence:** 5

**Summary:**

The paper introduces AUC-Contrastive Learning as a method to enable self-supervised learning even with a small batch size. This proposed method optimizes the model by making binary predictions on positive and negative samples, aiming to maximize the AUC score. Despite using a significantly smaller batch size, the method shows improved performance over existing contrastive approaches.

**Strengths:**

- The paper presents a contrastive learning technique based on the AUC score, effectively approximating a deterministic score. Its simplicity makes it easy to implement.

- Unlike conventional contrastive learning-based self-supervised approaches, the proposed method demonstrates strong robustness to batch size variations while also showing performance enhancements.

- Through extensive experiments, such as those involving various architectures, smaller datasets, and few-shot transfer, the paper proves the effectiveness of the method across different settings.

**Weaknesses:**

- The paper is generally not reader-friendly and lacks clarity. In several sections, terms are not defined or are vaguely explained. Specific issues include:
    - On the second page, the terms B and T are not defined, making comprehension difficult.
    - The introduction should provide a full explanation of AUC.
    - On the third page, the term N is not explained.
    - Figure 1 is distant from its related equation, making it hard to understand terms like A_1,2,3.
    - In Eq. 8, there's a lack of clarity on how 'a' and 'b' are replaced.
    - The algorithm section doesn’t exactly mention which loss function is being optimized.
    - The paper introduces each method name without specifying which previous work it refers to.

- The experimental results appear to be unfair, and several state-of-the-art methods are not reported.
    - In Table 1, many methods didn't use the multi-crop strategy, but the proposed method did. This gives the proposed method an unjust
 benefit as it would've seen many more images per epoch, leading to potentially improved results.
    - In Table 2, recent methods like DINO are not mentioned at all. DINO reported performance metrics (KNN: 72.8, Linear: 76.1 in 300 epochs) that are higher than the proposed method. The paper's claim that it outperforms several state-of-the-art methods is misleading, and a comprehensive comparison with all state-of-the-art methods is required.

**Questions:**

There needs to be a clearer explanation regarding how a(w) and b(w) are replaced in Equation 7 and what 'a' and 'b' represent in Equation 8.

---

> ### Author Response · Authors · 2023-11-17
>
> We are grateful to the reviewer for providing the critique which we aim to address and strengthen our contribution.
>
> **On the second page, the terms B and T are not defined, making comprehension difficult.**
>
> The terms $B$ and $T$ refer to the batch size of the training and the number of training steps. We merely use the common convention in referring to these quantities, in keeping with the larger context of the introduction. We shall update the manuscript in order to clarify further.
>
> **The introduction should provide a full explanation of AUC.**
>
> We chose to incorporate the complete definition and the methods used to optimize the metric in the method section with the intent of simplifying the introduction and motivations to the readers and avoid mathematical definitions and derivations in the early phases of the paper. We kindly direct the reviewer to section 3.2 for a deeper explanation of the AUC metric.
>
> **On the third page, the term N is not explained.**
>
> We were merely using the Big-O notation for time and space complexity wherein the term $N$ refers to the number of samples.
>
> **Figure 1 is distant from its related equation, making it hard to understand terms like A_1,2,3.**
>
> We apologize for the inconvenience in this regard. Figure 1 refers to the Equation 8 (next page). We shall clarify the reference in Section 3.3 which contains our main equation.
>
> **In Eq. 8, there's a lack of clarity on how 'a' and 'b' are replaced.**
>
> In Equation 8, the parameters $a$ and $b$ undergo optimization by the network, akin to AUC maximization (referenced as Equation 4 in [1]). Notably, we opt to fix $a$ and optimize $b$, a decision grounded in empirical considerations. This choice is made to enhance the stability of the training process, leveraging the network's greater ease of optimization, as articulated in the manuscript.
>
> **The algorithm section doesn’t exactly mention which loss function is being optimized.**
>
> We thank the reviewer for pointing out this oversight to us. The algorithm optimizes Equation 8, which we have now corrected in the manuscript.
>
> **The paper introduces each method name without specifying which previous work it refers to.**
>
> We express gratitude to the reviewer for their observation. Citations to the methods in Tables 2 and 3 are embedded in the manuscript's text. To enhance clarity, we have revised these citations, now incorporating the names of the methods.
>
> **In Table 1, many methods didn't use the multi-crop strategy, but the proposed method did. This gives the proposed method an unjust benefit as it would've seen many more images per epoch, leading to potentially improved results.**
>
> We express our gratitude to the reviewer for highlighting this observation. In the subsequent analysis, we aim to showcase our method's performance without employing the multi-crop strategy. Specifically, we compare our approach against DINO, omitting the multi-crop strategy, utilizing the ViT-S backbone, and pretraining on ImageNet for 100 epochs with a batch size of 128. Notably, DINO employed a batch size of 1024. The results are extracted from Table 8 in DINO and Table 6 in our manuscript.
>
> |Method	|Batch Size	|Linear Acc@1|
> |:----|:----:|:----:|
> |DINO	|1024	|67.8|
> |MoCo-v3 |512		|62.3|
> |Ours	|256		|**67.9**|
>
> Additionally, we kindly direct the reviewer's attention to the results presented in the manuscript, excluding Tables 1, 2, and 4. These results are obtained without the use of the multi-crop augmentation strategy, following the standard augmentation method [2]. We compare against all methods listed in the main ImageNet comparisons in these tables. We assert that our method consistently demonstrates state-of-the-art performance without relying on the multi-crop strategy, as evident in multiple manuscript results. Furthermore, in the common response table, we present the outcome of pretraining ResNet-50 on ImageNet using our method for 650 epochs with the multi-crop strategy. Notably, the other figures in the table align with accepted work from DINO ([3], Table 2), encompassing contributions with and without the use of the multi-crop strategy.
>
>
> [1] Yuan, Zhuoning, et al. "Large-scale robust deep auc maximization: A new surrogate loss and empirical studies on medical image classification." Proceedings of the IEEE/CVF International Conference on Computer Vision. 2021.
>
> [2] Chen, Ting, et al. "A simple framework for contrastive learning of visual representations." International conference on machine learning. PMLR, 2020.
>
> [3] Caron, Mathilde, et al. "Emerging properties in self-supervised vision transformers." Proceedings of the IEEE/CVF international conference on computer vision. 2021.

---

> ### Author Response · Authors · 2023-11-17
>
> **In Table 2, recent methods like DINO are not mentioned at all. DINO reported performance metrics (KNN: 72.8, Linear: 76.1 in 300 epochs) that are higher than the proposed method. The paper's claim that it outperforms several state-of-the-art methods is misleading, and a comprehensive comparison with all state-of-the-art methods is required.**
>
> We express gratitude to the reviewer for this feedback and we will add and clarify this result in revision. It is pertinent to note that extensive efforts have been made to encompass well-known state-of-the-art methods in our comparisons, as demonstrated in Table 1, where DINO is included. We outperform DINO in Tables 1 and 3. Additionally, we hope the reviewer can kindly also consider our newer results mentioned in the common response section.

---

> ### Author Response · Authors · 2023-11-22
> **Gentle reminder**
>
> Dear reviewer nw59,
>
> We express our gratitude for your dedicated efforts in reviewing and providing valuable suggestions. Despite the prolonged discussion period, we have yet to receive feedback on your response. Kindly note that we have concluded the pretraining of Resnet-50, as outlined in the main response, and have incorporated the necessary updates into the PDF (appendix, Table 6), including the suggested adjustments to method names and references. We would really appreciate it if you can let us know if our response resolves your concerns. Additionally, please inform us of any lingering concerns you may have.
>
> Best regards, Authors

---

### Official Review · Reviewer_vuAt · 2023-10-31

**Soundness:** 2 fair
**Presentation:** 3 good
**Contribution:** 2 fair
**Rating:** 6
**Confidence:** 2

**Summary:**

This paper introduces AUC-CL, a new approach to self-supervised contrastive learning that seeks to mitigate the dependencies on large batch sizes required by many existing contrastive methods. By integrating the contrastive objective within the AUC-maximization framework, the method prioritizes improving the binary prediction differences between positive and negative samples. This unique approach offers unbiased stochastic gradients during optimization, suggesting a higher resilience to batch size variations. Experimental results indicate that the AUC-Contrastive Learning, even with smaller batch sizes like 256, can rival or surpass performances of other methods requiring considerably larger batch sizes.

**Strengths:**

1. This paper addresses an important issue in contrast learning: model performance is particularly sensitive to batch size, preferring large batch size.

2. The theoretical proof of the design in this paper is very detailed.

**Weaknesses:**

1. There are non-contrastive learning SSL methods (e.g., SimSiam) that are also less sensitive to batch size. Considering those methods, the novelty of this paper might be a concern.

2. There appears to be a conflict between the experimental results in the table and the figure.

**Questions:**

1. The contrastive learning methods (e.g., SimCLR and MoCo) are sensitive to batch size and prefer larger batch size. However, there are some self-supervised leanring methods such as SimSiam, which do not have the concept of "positive" and "negative" pairs and are more robust to smaller batch sizes. Considering those methods, the contribution of this paper is diminished.

2. In Table 3, the method proposed in this paper outperforms SimSiam in Cifar-10 dataset, with a 3.1% higher accuracy. However, if we take a look at the Figure 3, the accuracy gap between them is very small, especially in late training stage (after 400 epochs). Could you explain why there appears to be a conflict between these two results?

3. As shown in Figure 3, the accuracy advantage of the proposed method gradually disappear as training proceeds, so it actually just converges faster at early training stage. Moreover, according to the convergence trend in Figure 3, if we train the model for 1,000 epochs for complete convergence as stated in Table 3, the proposed method will not even have an advantage convergence speed.

---

> ### Author Response · Authors · 2023-11-17
>
> **The contrastive learning methods (e.g., SimCLR and MoCo) are sensitive to batch size and prefer larger batch size. However, there are some self-supervised leanring methods such as SimSiam, which do not have the concept of "positive" and "negative" pairs and are more robust to smaller batch sizes. Considering those methods, the contribution of this paper is diminished.**
>
> We express our appreciation to the reviewer for their insightful feedback. In response, we respectfully direct the reviewer's attention to our comparative analysis, presented in Tables 1 and 3, which contrasts our method, SimSiam, with several others claiming robustness to batch size. Notably, these methods, namely BYOL [1], Barlow Twins [2], Zero-CL [3], W-MSE [4], ARB [5] and DINO [6], are all negative-free techniques. Our work demonstrates superior performance on the specified datasets at a significantly smaller batch size, showcasing heightened robustness.
>
> It is imperative to highlight that while these methods are negative-free, our noteworthy advantage lies in achieving **superior performance with a considerably reduced batch size**. This distinction enables us to execute training on limited computational resources, underscoring the unique contribution of our work. Furthermore, we would like to emphasize that contrastive learning, incorporating negatives, remains an active and evolving research domain despite the advancements in non-contrastive methods. Some exemplary and recent approaches in this area include [7, 8, 9].
>
> **In Table 3, the method proposed in this paper outperforms SimSiam in Cifar-10 dataset, with a 3.1% higher accuracy. However, if we take a look at the Figure 3, the accuracy gap between them is very small, especially in late training stage (after 400 epochs). Could you explain why there appears to be a conflict between these two results?**
>
> We are afraid that the reviewer may be mistaken in this evaluation. Table 3 lists the *linear evaluation* results after pretraining for 500 epochs using a batch size of 128, while Figure 3 lists the *k-NN* accuracy (Acc @1) for the mentioned methods.
> It is crucial to highlight that, despite the close proximity in k-NN accuracy, the features derived from our pretraining method exhibit a greater informativeness for downstream tasks, as is evinced by the linear evaluation performance given in Table 3.
>
>
> [1] Grill, Jean-Bastien, et al. "Bootstrap your own latent-a new approach to self-supervised learning." Advances in neural information processing systems 33 (2020): 21271-21284.
>
> [2] Zbontar, Jure, et al. "Barlow twins: Self-supervised learning via redundancy reduction." International Conference on Machine Learning. PMLR, 2021.
>
> [3] Zhang, Shaofeng, et al. "Zero-cl: Instance and feature decorrelation for negative-free symmetric contrastive learning." International Conference on Learning Representations. 2021.
>
> [4]  Ermolov, Aleksandr, et al. "Whitening for self-supervised representation learning." International Conference on Machine Learning. PMLR, 2021.
>
> [5] Zhang, Shaofeng, et al. "Align representations with base: A new approach to self-supervised learning." Proceedings of the IEEE/CVF Conference on Computer Vision and Pattern Recognition. 2022.
>
> [6] Caron, Mathilde, et al. "Emerging properties in self-supervised vision transformers." Proceedings of the IEEE/CVF international conference on computer vision. 2021.
>
> [7] Jiang, Qian, et al. "Understanding and constructing latent modality structures in multi-modal representation learning." Proceedings of the IEEE/CVF Conference on Computer Vision and Pattern Recognition. 2023.
>
> [8] Johnson, Daniel D., Ayoub El Hanchi, and Chris J. Maddison. "Contrastive learning can find an optimal basis for approximately view-invariant functions." arXiv preprint arXiv:2210.01883 (2022).
>
> [9] Hu, Tianyang, et al. "Your contrastive learning is secretly doing stochastic neighbor embedding." arXiv preprint arXiv:2205.14814 (2022).

---

> > ### Author Response · Authors · 2023-11-17
> >
> > **As shown in Figure 3, the accuracy advantage of the proposed method gradually disappear as training proceeds, so it actually just converges faster at early training stage. Moreover, according to the convergence trend in Figure 3, if we train the model for 1,000 epochs for complete convergence as stated in Table 3, the proposed method will not even have an advantage convergence speed.**
> >
> > We extend our gratitude to the reviewer for the feedback. Firstly we direct the reviewer's attention to the common response, where we illustrate the results of our method upon training for a higher number of epochs on ImageNet using ResNet-50. However, it is essential to underscore that our method's comparison in Table 3 involves pretraining for half the epochs (500) compared to other methods (1000 epochs). This deliberate choice aims to showcase our method's distinct advantage in early convergence. Remarkably, our method not only converges swiftly but attains a superior optimum, as evident from its performance in linear evaluation. This contrasts with the common trade-off between performance and convergence observed in many methods. We contend that this characteristic empowers users to conduct resource-efficient self-supervised training. It allows for training a highly competent model with a smaller batch size and fewer pretraining epochs, rendering the method versatile and effective. This, in turn, contributes to savings, addressing the contemporary concern of large model footprints. However, in the subsequent table, we present results after pretraining our method for 1000 epochs, utilizing the ResNet-18 backbone on the Cifar-10 dataset. Standard linear evaluation follows the specified protocol and parameters in the manuscript.
> >
> > |Method	|Batch Size	|Linear (Acc@1)|
> > |:----|:----:|:----:|
> > |SimSiam	|256	|90.5|
> > |Barlow Twins	|256	|92.1|
> > |ARB		|256	|92.2|
> > |BYOL	|256	|92.6|
> > |MoCo-v3|256	|93.1|
> > |Ours	|128	|**93.9**|
> >
> >
> > We include the performances of the best performing models from Table 3 here and SimSiam. Notably, the performance benefits are further enhanced with prolonged training. We emphasize that our method, algorithmically and without architectural modifications, facilitates training models capable of surpassing state-of-the-art performances using practical and modest resources. This departure from the norm in modern AI, especially in self-supervised learning, challenges the prevalent use of larger models and batch sizes for optimal performance.

---

> > > ### Comment · Reviewer_vuAt · 2023-11-23
> > > **Response to Authors**
> > >
> > > Thanks for the clarification and additional results. I agree this work is an effective method to make contrastive learning more robust to batch size. I have raised my score to 6.
> > >
> > > However, I am also aware that the authors claim to accelerate the model convergence. There have already been some works [1][2][3] targeting model convergence acceleration in self-supervised learning, which need to be discussed. Specifically, [1] claims to accelerate contrastive learning 8 times, which is an impressive number.
> > >
> > > [1] Ci, Yuanzheng, et al. "Fast-MoCo: Boost momentum-based contrastive learning with combinatorial patches." European Conference on Computer Vision. Cham: Springer Nature Switzerland, 2022.
> > >
> > > [2] Addepalli, Sravanti, et al. "Towards Efficient and Effective Self-Supervised Learning of Visual Representations." European Conference on Computer Vision. Cham: Springer Nature Switzerland, 2022.
> > >
> > > [3] Koçyiğit, Mustafa Taha, Timothy M. Hospedales, and Hakan Bilen. "Accelerating Self-Supervised Learning via Efficient Training Strategies." Proceedings of the IEEE/CVF Winter Conference on Applications of Computer Vision. 2023.

---

> ### Author Response · Authors · 2023-11-22
> **Gentle reminder**
>
> Dear reviewer vuAt,
>
> We express our gratitude for your dedicated efforts in reviewing and providing valuable suggestions. Despite the prolonged discussion period, we have yet to receive feedback on your response. Kindly note that we have concluded the pretraining of Resnet-50, as outlined in the main response, and have incorporated the necessary updates into the PDF (Appendix, Table 6), including minor formatting adjustments. We would really appreciate it if you can let us know if our response resolves your concerns. Additionally, please inform us of any lingering concerns you may have.
>
> Best regards, Authors

---

> ### Author Response · Authors · 2023-11-23
> **Thank you so much for your feedback**
>
> Dear reviewer vuAt,
>
> Thank you so much for your response! We shall duly incorporate a discussion drawing parallels between our method and the suggested approaches.
>
> Thanking you,
> Authors

---

### Author Response · Authors · 2023-11-17
**Common Response**

We express our sincere gratitude to the reviewers for their invaluable critiques and thoughtful responses to our work. We are pleased that the reviewers find our work significantly engaging. Common concerns raised by the reviewers pertain to the absolute performance of the method. To address this, we present results obtained by training our method for 650 epochs using ResNet-50 on ImageNet, using the same batch size of 256 as in the manuscript. Kindly note that we are still waiting for the training to complete 800 epochs and will add the additional result to our main comparison in Table 1. We have released the evaluation logs in the codebase for this manuscript.

Edit: Kindly note that we have completed the procedure for pretraining for 800 epochs and linear evaluation and incorporated the result in the table below and the manuscript (appendix, Table 6).

|Method | Epochs | Linear Acc@1|
|:----|:----:|:----:|
|SimCLR	|800		|69.1|
|MoCov2	|800		|71.1|
|InfoMin 	|800		|73.0|
|Barlow Twins 	|800		|73.2|
|OBOW	|800		|73.8|
|BYOL	|800		|74.4|
|DCv2	 	|800		|*75.2*|
|SwAV	|800		|*75.3*|
|DINO	|800		|*75.3*|
|Ours		|400		|**73.5**|
|Ours		|650		|**75.0**|
|Ours		|800		|**75.5**|

Additionally, the following are some of the clarifications.
1. Our motivation was not aimed at overcoming the SOTA scores but to provide a resource efficient method allowing independent researchers to replicably train a self supervised framework using an easily implementable loss function that enables faster and better convergence that is demonstrably unbiased by batch size.
2. We understand that a large proportion of the work in self supervised learning (SSL) evaluates their methods on very large batch sizes, often as high as or higher than 8192 where accessibility to hardware capable of training on such batch sizes is crucial. For example, MoCo-v3 uses the most powerful TPU servers for large batch-size experiments, which is often prohibitive to most researchers. We believe that in this era of AI revolution, the rate of growth of datasets is projected to be far higher than that of the accessibility and innovation in superior hardware. We posit that our work deems to make accessible research in SSL to independent researchers where access to such hardware is limited and propose a principled approach to this impediment, that is conceptually simple and empirically robust.
3. We argue that absolute performance may not be the sole criterion for a contribution's validity. In this context, we bring the reviewers’ attention to several notable and recent contributions [1, 2, 3, 4] that demonstrate an alternative method with specific benefits that enable SSL, **without aiming for or achieving an absolute SOTA performance**.
4. We have diligently adhered to widely accepted practices and architectures within the realm of self-supervised learning. Notable contributions include [1, 2, 3, 4, 5, 6, 7] which use the ResNet-50 architecture as one of their primary training backbones. Comparisons of our method against contributions that use much larger and newer architectures may be unfair.
5. We appreciate reviewers' feedback on manuscript formatting, and relevant adjustments have been implemented.



[1]  Ermolov, Aleksandr, et al. "Whitening for self-supervised representation learning." International Conference on Machine Learning. PMLR, 2021.

[2] Zhang, Shaofeng, et al. "Zero-cl: Instance and feature decorrelation for negative-free symmetric contrastive learning." International Conference on Learning Representations. 2021.

[3] Ni, Renkun, et al. "The close relationship between contrastive learning and meta-learning." International Conference on Learning Representations. 2021.

[4] Zhang, Shaofeng, et al. "Align representations with base: A new approach to self-supervised learning." Proceedings of the IEEE/CVF Conference on Computer Vision and Pattern Recognition. 2022.

[5] Zbontar, Jure, et al. "Barlow twins: Self-supervised learning via redundancy reduction." International Conference on Machine Learning. PMLR, 2021.

[6] Grill, Jean-Bastien, et al. "Bootstrap your own latent-a new approach to self-supervised learning." Advances in neural information processing systems 33 (2020): 21271-21284.

[7] Chen, Ting, et al. "A simple framework for contrastive learning of visual representations." International conference on machine learning. PMLR, 2020.

---

### Meta-Review · Area_Chair_nd9b · 2023-12-18

**Metareview:**

The paper introduces a new loss function for contrastive learning that is more robust to batch sizes - as a result it could be feasible to perform contrastive learning with small batch sizes. Requiring large batch size is one of the biggest compute hurdles and this paper makes a promising contribution. While I think this paper should be accepted, there are clear avenues for improvement as noted in the reviews: better positioning with respect to baselines, carefully controlling various aspects such as augmentations when comparing across methods (this is especially important since the paper doesn't claim to get SOTA but shows on a smaller scale that one can match previous methods but with much smaller batch sizes)

**Justification For Why Not Higher Score:**

The experimental setup could be more sound and consistent

**Justification For Why Not Lower Score:**

Method is interesting, addresses an important problem, is likely to scale well because it's simple

---

### Decision · Program_Chairs · 2024-01-16

Accept (poster)